

# Distinct gene expression and secondary metabolite profiles in *suppressor of prosystemin-mediated responses2 (spr2)* tomato mutants having impaired mycorrhizal colonization

Kena Casarrubias-Castillo[1], Josaphat M. Montero-Vargas[2], Nicole Dabdoub-González[3], Robert Winkler[4], Norma A. Martinez-Gallardo[4], Julia Zañudo-Hernández[1], Hamlet Avilés-Arnaut[3] and John P. Délano-Frier[4]

[1] Centro Universitario de Ciencias Biológicas y Agropecuarias, Universidad de Guadalajara, Zapopan, Jalisco, Mexico
[2] Departamento de Investigación en Inmunogenética y Alergia, Instituto Nacional de Enfermedades Respiratorias "Ismael Cosío Villegas", Mexico City, Mexico
[3] Instituto de Biotecnología de la Facultad de Ciencias Biológicas, Universidad Autónoma de Nuevo León, Nicolas de los Garza, Nuevo Leon, Mexico
[4] Departamento de Biotecnología y Bioquímica, Centro de Investigación y de Estudios Avanzados del IPN-Unidad Irapuato, Irapuato, Guanajuato, México

Corresponding authors
Hamlet Avilés-Arnaut,
hamlet.avilesarn@uanl.edu.mx
John P. Délano-Frier,
john.delano@cinvestav.mx

## ABSTRACT

Arbuscular mycorrhizal fungi (AMF) colonization, sampled at 32–50 days post-inoculation (dpi), was significantly reduced in *suppressor of prosystemin-mediated responses2 (spr2)* mutant tomato plants impaired in the ω−3 *FATTY ACID DESATURASE7* (*FAD7*) gene that limits the generation of linolenic acid and, consequently, the wound-responsive jasmonic acid (JA) burst. Contrary to wild-type (WT) plants, JA levels in root and leaves of *spr2* mutants remained unchanged in response to AMF colonization, further supporting its regulatory role in the AM symbiosis. Decreased AMF colonization in *spr2* plants was also linked to alterations associated with a disrupted FAD7 function, such as enhanced salicylic acid (SA) levels and SA-related defense gene expression and a reduction in fatty acid content in both mycorrhizal *spr2* roots and leaves. Transcriptomic data revealed that lower mycorrhizal colonization efficiency in *spr2* mutants coincided with the modified expression of key genes controlling gibberellin and ethylene signaling, brassinosteroid, ethylene, apocarotenoid and phenylpropanoid synthesis, and the wound response. Targeted metabolomic analysis, performed at 45 dpi, revealed augmented contents of L-threonic acid and DL-malic acid in colonized *spr2* roots which suggested unfavorable conditions for AMF colonization. Additionally, time- and genotype-dependent changes in root steroid glycoalkaloid levels, including tomatine, suggested that these metabolites might positively regulate the AM symbiosis in tomato. Untargeted metabolomic analysis demonstrated that the tomato root metabolomes were distinctly affected by genotype, mycorrhizal colonization and colonization time. In conclusion, reduced AMF colonization efficiency in *spr2* mutants is probably caused by multiple and interconnected JA-dependent and independent gene expression and metabolomic alterations.

## INTRODUCTION

The roots of the majority of higher plants, and many other host plants including pteridophytes, a number of mosses, lycopods, and psilotales are associated symbiotically with biotrophic and aseptate filamentous fungi of the Glomeromycota phylum, the so-called arbuscular mycorrhizal fungi (AMF) (*Redecker & Raab, 2006*; *Schüssler, Schwarzott & Walker, 2001*; *Van der Heijden et al., 2015*). An important feature of the AM symbiosis is the nutrient exchange between both partners, which occurs within individual cortical cells, where AMF form highly branched hyphae, called arbuscules, surrounded by a plant derived periarbuscular membrane. This structure has a unique transport protein distribution designed to deliver mineral nutrients to the cortical cell in exchange for the 4–20% of the photosynthates allocated to the roots and subsequently transferred to the fungal symbiont (*Gutjahr & Parniske, 2013*; *Bhandari & Garg, 2017*; *Rich et al., 2017*).

The mineral exchange principally involves phosphorus (P), although zinc and copper are also thought to be mobilized. AMF substantially enhance P availability to plants considering that this element is mostly found as orthophosphate ions in the soil, which become very poorly mobile in the presence of $Ca^{2+}$, $Fe^{3+}$ and $Al^{3+}$ (*Wang et al., 2017*). P uptake relies on phosphate transporters belonging to the phosphate transporter 1 (Pht1) family (*Rausch et al., 2001*). AMF are also believed to represent a significant route of nitrogen (N) uptake by the plants (*Délano-Frier & Tejeda-Sartorius, 2008*). In addition to improving plant nutrition and water uptake, root colonization by AMF confers fitness benefits to the host plants. Among these are enhanced root growth and significant changes in root architecture that increase their surface area in the soil and enhance water and nutrient acquisition by the plants. Other benefits include increased reproductive success and/ or tolerance to (a)biotic stresses. Several of the above effects are finely regulated by phytohormones, via a limitedly understood and highly context dependent interaction (*Gutjahr, 2014*; *Selosse, Bessis & Pozo, 2014*; *Pozo et al., 2015*; *Lenoir, Fontaine & Sahraoui, 2016*; *Bedini et al., 2018*; *Liao et al., 2018*; *Cabral et al., 2019*; *Evelin et al., 2019*).

Arbuscular mycorrhizal fungi colonization has been found to profoundly alter gene expression in roots of several plants species (*Guimil et al., 2005*; *Fiorilli et al., 2009*; *Guether et al., 2009*; *Hogekamp & Küster, 2013*; *Groten et al., 2015*), whereas a number of large-scale gene expression studies have explored the systemic effects that AMF root colonization has in plant aerial tissues (*Guimil et al., 2005*; *Cervantes-Gamez et al., 2016*; *Adolfsson et al., 2017*; *Wang et al., 2018a*). Changes in root and leaf metabolite levels in response to AMF colonization, have been also reported, particularly affecting amino acid (aa), carbohydrate and organic acid contents (*Schweiger et al., 2014*; *Rivero et al., 2015*; *Schweiger & Müller, 2015*; *French, 2017*; *Bedini et al., 2018*; *Wang et al., 2018a*).

Jasmonic acid (JA), its methyl ester (MeJA) and its aa conjugates such as JA-isoleucine, collectively referred as jasmonates, are involved in multiple plant developmental and growth processes, activation of secondary metabolism and defense responses against

insects and pathogens in various plant species (*Wasternack & Hause, 2013*; *Erb, Meldau & Howe, 2012*; *Ahmad et al., 2016*; *Larrieu & Vernou, 2016*). JA is biosynthesized through the octadecanoid pathway, being the final product of a series of reactions initiated by the peroxidation of linolenic acid, a C18:3 unsaturated fatty acid (*Schaller, Schaller & Stintzi, 2005*). Recent models indicate that JA is the long-distance systemic wound signal in tomato. Wound-induced systemic signaling appears to be facilitated by a positive amplification loop in which systemin, a wound related bioactive peptide signal, is considered to play a crucial role (*Schilmiller & Howe, 2005*; *Sun, Jiang & Li, 2011*). Experimental evidence gathered so far portrays a contradictory role for JA in the regulation of the mycorrhizal symbiosis (*Liao et al., 2018*; *Pozo et al., 2015*). While some findings have led to the proposal of several mechanisms designed to explain how JA might positively regulate the AM symbiosis (*Isayenkov et al., 2005*; *Stumpe et al., 2005*; *Tejeda-Sartorius, Martínez de la Vega & Délano-Frier, 2008*), others have reported reduced or null effects on AMF colonization in tomato plants treated with exogenous MeJA or in tomato and rice mutants unable to sense or synthesize JA (*Herrera-Medina et al., 2008*; *Gutjahr & Paszkowski, 2009*; *Gutjahr et al., 2015*; *Bedini et al., 2018*). Additionally, it has been suggested that JA signaling might differentially affect AMF colonization and development by performing different roles in early and late stages of colonization (*Foo et al., 2013*).

Additional findings, derived from the use of the *spr2* tomato mutant as an experimental model (*Howe & Ryan, 1999*) are reported in this work. The *spr2* tomato mutant is affected in a chloroplast FAD7 fatty acid desaturase that catalyzes the ω3 desaturation of linoleic acid (C18:2) to linolenic acid, JA's biosynthetic precursor. It is also impaired in the wound-induced accumulation of JA and also in the developmentally regulated JA burst in flowers (*Li et al., 2002*, *2003*). In addition, the disrupted function of FAD7 has been found to enhance plant defense against aphids due to increased SA levels and SA-related defense gene expression (*Avila et al., 2012*). The precise mechanism responsible for this effect remains undefined, although direct and indirect FAD7-mediated modulation of SA biosynthesis in the chloroplast has been proposed, the latter via the accumulation of reactive oxygen species (ROS).

The results derived from the present study provide congruent arguments to explain how mutant plants negatively affected in the wound-induced accumulation of JA and the subsequent wound-response, are also disrupted in their ability to establish an efficient AM symbiosis. For this purpose, wild-type (WT) and *spr2* plants were analyzed to measure changes in gene expression and metabolic profiles in response to AMF colonization at various sampling points. The results obtained suggest that deficient mycorrhization in *spr2* tomato roots was caused by several factors. Many could be related to physiological changes directly related to the *FAD7* mutation, such as increased SA levels and activation of SA-related gene expression in addition to an alteration in fatty acid composition and abundance in leaves and roots. Indirect effects, occurring as a consequence of the mutant's inability to undergo mycorrhizal-induced JA accumulation in roots, could have contributed to alter the expression of key genes known to influence AMF colonization, via phytohormone biosynthesis and signaling, in addition to the regulation of the wound-response and of secondary metabolism. In this respect, differential time- and

genotype-dependent changes in tomatine, its biosynthetic precursors and catabolic products, in mycorrhizal WT and *spr2* roots, suggested a role of this defense-related metabolite in the regulation of the AM symbiosis. These results are discussed further in the context of the numerous and interconnected mechanisms that control colonization of plants by AMF.

## MATERIALS AND METHODS

### Plant growth and AMF inoculation

Seeds of wild-type (WT) tomato (*Solanum lycopersicum* L. cv. Castlemart) and *spr2* mutant plants were surface-sterilized by soaking in a 70% ethanol solution for 60 s, with a 20% household bleach solution (5% w/v sodium hypochlorite) for 5 min, and then rinsed three times with sterile water. All seeds were germinated in a sterile soil mixture constituted by equal parts of sand and loam which was autoclaved six times. One-week-old seedlings were removed and transplanted to 1.3 L pots (one plant per pot, ten plants per genotype) containing the same sterilized soil mixture. At the time of transplanting, ten plants per genotype were inoculated with 3 g of a soil-based (1:1 sand-loam) inoculum containing ca. 100 AMF spores per g. In the first (E1) of a series of three experiments, performed between April and May 2016, the plant's inoculum was *Rhizophagus irregularis* (*Biofertilizante*, INIFAP, México), whereas in two subsequent ones (E2 and E3, performed between early April-late May and late May-early July 2019, respectively), a consortium of six AMF species, that is, *Glomus fasciculatus, G. constrictum, G. tortuosum, G. geosporum, Gigaspora margarita*, and *Acaulospora scrobicurata* (MycoRacine_VA, MycoBiosfera, México), was used. Similar number of control plants were supplied with 3 g of sterilized soil mixture only. The inoculated and control plants were watered 3 times per week and fertilized once a week with a Long Asthon solution in which the P content was reduced to 7 μM (75% lower than that in the full strength solution), until harvest. They were kept in a growth room with a 16 h/ 8 h light/ dark photoperiod at 27 °C (light) and 23 °C (dark) with an illumination of approximately 250 μmol m$^{-2}$ s$^{-1}$. Roots were harvested at different time points: at 50 (E1), 45 (E2) or 32 (E3) days post-inoculation (dpi). The root system was split lengthways at harvest: one half was stained to evaluate mycorrhizal colonization, whereas the remaining root and leaf tissues were frozen, ground in liquid nitrogen, stored at −80 °C until required for further analysis.

### Estimation of AMF root colonization

To evaluate mycorrhizal colonization, root fragments of control and mycorrhized plants (120 per genotype) were stained with trypan blue (*Phillips & Hayman, 1970*) and observed with a light microscope. AMF colonization was determined in the three independent experiments using two methods: E1 was analyzed as described by *Trouvelot, Kough & Gianinazzi-Pearson (1986)* using MYCOCALC software (www2.dijon.inra.fr/mychintec/Mycocalc-prg/download.html), while E2 and E3 were analyzed using the magnified intersections method described by *McGonigle et al. (1990)*. Three colonization parameters were analyzed in E1: frequency of mycorrhization (F%), representing the percentage of root segments showing internal colonization; intensity of mycorrhization (M%),

the average percentage of colonization of root segments, and arbuscule abundance (A%), the percentage of arbuscules in the whole root system. In E2 and E3, AMF colonization levels were determined by quantifying the proportion of root length segments containing arbuscules, vesicles and/or hyphae, each representing the arbuscular (A), vesicular (V) and hyphal colonization (H) levels, respectively.

Additionally, physiological parameters such as plant height and changes in chlorophyll content were measured during the duration of the E2 experiment. In addition, maximal photochemical efficiency (*Fv/Fm*) and the photosynthetic potential Index (PI$_{abs}$) were measured as indirect indicators of the quantum efficiency of photosystem II. Chlorophyll content was measured using a CCM-200 plus Chlorophyll Content Meter (Opti-Sciences Inc., Hudson, NH, USA). *Fv/Fm* and PI$_{abs}$ were determined from data recorded with a portable Pocket PEA chlorophyll fluorometer (Hansatech Instruments Ltd.; Norfolk, UK). All photosynthesis-related measurements were consistently done at noon.

## Determination of jasmonic acid (JA), salicylic acid (SA) and SA conjugates

Jasmonic acid levels were determined by GC–MS using the methodology described by *Muller & Brodschelm (1994)*. Free SA and SA conjugates were also analyzed by GC–MS, as described by *Malamy, Henning & Klessig (1992)*.

## Targeted metabolite profiling and direct fatty acid (FA) analysis by gas chromatography–mass spectrometry

The GC–MS targeted metabolite analysis of leaf and root samples was performed according to *Eloh et al. (2016)*. In situ FA analysis was done as described by *Park & Goind (1994)*, with slight modifications. These consisted of increasing the 10 min-each methanolysis and methylation steps to 60 and 30 min, respectively. Raw GC–MS data are available from Zenodo (https://zenodo.org/), DOI 10.5281/zenodo.3560965.

## Sample preparation for untargeted metabolomic analyses

Frozen tomato roots of control and mycorrhizal plants, respectively, were lyophilized and finely ground in a Mixer Mill MM 400 (Retsch GmbH, Haan, Germany) for 12 s at 30 Hz. Subsequently, 25 mg of plant powder was extracted with 1 mL of an aqueous methanol-formic acid solution (75% v/v and 0.15% v/v, respectively). The mixture was sonicated for 15 min in a water bath at maximum frequency and centrifuged at $10,000 \times g$ for 10 min at 4 °C. The supernatants were filtered through a 0.22 μm syringe filter prior to analysis by mass spectrometry (see below). All samples were freshly prepared in triplicate.

## Metabolic fingerprinting of tomato root and leaf extracts by mass spectrometry

For non-targeted metabolite profiling, tomato root methanolic extracts were analyzed by direct liquid introduction electrospray ionization/mass spectrometry (DLI–ESI–MS) as described before (*Montero-Vargas et al., 2013*). Measurements were performed using an ion trap mass spectrometer LCQ-Fleet (Thermo Scientific, Waltham, MA, USA) in

positive mode at a flow rate of 10 μL min$^{-1}$. Mass spectra were acquired in continuous mode in a range of 50–1300 m/z. The scan time was set to 300 ms with three microscans repeated ten times. The instrument settings were: 3.9 kV source voltage, 35 V capillary voltage, 330 °C capillary temperature, 80 V tube lens voltage, 15 arbitrary units (AU) of nitrogen sheath gas and 20 AU of auxiliary gas. Additionally, samples were analyzed to detect specific ions corresponding to α-tomatine, its biosynthetic precursors and catabolic products (*Montero-Vargas et al., 2018*).

## Raw data processing and data analysis

Raw mass spectrometry data were converted to .mzML format with MSConvert (*Chambers et al., 2012*). The mass spectrum data were processed in R (http://www.rproject.org) using the package MALDIquant version 1.15 (*Gibb & Strimmer, 2012*) programed to execute the following tasks: .mzML data import, summary of all scans of each spectrum, smoothing by a Savitzky-Golay filter, peak alignment and peak selection detection. Raw data are available from Zenodo (https://zenodo.org/), DOI: 10.5281/zenodo.3560965. Finally, a comparison matrix (m/z values for columns and file names for rows) with the intensity of peaks were exported in *csv* format for statistical analysis, applying binning with a bin width of 1 m/z and intensity-based normalization. A total of 367, from more than 1,000 ion signals, were used after initial data cleaning steps.

Unsupervised techniques were employed to explore the effect of mycorrhizal on the metabolic fingerprints of the genotypes employed. These consisted of a hierarchical clustering (HCA) and a principal component analysis (PCA), both implemented in ClustVis (*Metsalu & Vilo, 2015*). For multiple comparison of the means a Tukey's Honest Significant Difference (HSD) was applied with a confidence interval of 95%. When the normality of the data was not achieved, the Kruskal–Wallis test was applied, considering $p$ values ≤ 0.05 as significant.

For putative metabolite identification, a homebuilt metabolite database for tomato was created based on previous reports (*Moco et al., 2006*; *Itkin et al., 2011*; *Caprioli et al., 2015*) and the SolCyc database (http://solcyc.solgenomics.net/). Subsequently, automatic matching of the m/z list was performed employing the software SpiderMass (*Winkler, 2015*).

## Extraction of total RNA, cDNA preparation and qPCR analysis

Total RNA was extracted from 100 to 500 mg of frozen root tissues with the Trizol reagent (Invitrogen, Carlsbad, CA, USA), according to the manufacturer's instructions, with modifications. These consisted of the addition of a salt solution (sodium citrate 0.8 M + 1.2 M NaCl) during precipitation in a 1: 1 v/v ratio with isopropanol and further purification with LiCl (8 M) for 1 h at 4 °C. All RNA samples were analyzed by formaldehyde agarose gel electrophoresis and visual inspection of the ribosomal RNA bands upon ethidium bromide staining. Total RNA samples (4 μg) were reverse-transcribed to generate the first-strand cDNA using an oligo dT20 primer and 200 units of SuperScript II reverse transcriptase (Invitrogen, Carlsbad, CA, USA). The cDNA employed for the qRT-PCR assays was initially prepared from 4 μg total RNA. It was then diluted 5-fold in sterile deionized-distilled (dd) water prior to qRT-PCR. Amplifications were
performed using SYBR Green detection chemistry and run in triplicate in 96-well reaction plates with the CFX96 Touch Real-Time PCR Detection System (Bio-Rad, Hercules, CA, USA). Reactions were prepared in a total volume of 20 µl containing: 2 µl of template, 2 µl of each amplification primer (2 µM), 8 µl of iQ SYBR Green supermix (Bio-Rad, Hercules, CA, USA) and 6 µl of sterile dd water. Quantitative real-time PCR was performed in triplicate for each sample using the primers listed in the Table S1. Primers were designed for each gene, based on cDNA sequences derived from the tomato genome (Sol Genomics Network; *Mueller et al., 2005*). Primer design was performed using DNA calculator software (Sigma-Aldrich, St. Louis, MO, USA). The following protocol was followed for all qRT-PCR runs: 15 min at 95 °C to activate the JumpStart Taq Polymerase (Sigma-Aldrich, St. Louis, MO, USA), followed by 40 cycles of denaturation at 95 °C for 15 s and annealing at 60 °C for 1 min. Slow amplifications requiring an excess of 32 cycles were not considered for analysis. The specificity of the amplicons was verified by melting curve analysis after 40 cycles. Baseline and threshold cycles (Ct) were automatically determined using CFX Manager Software version 2.1. PCR efficiencies for all genes tested were greater than 95%. The effect of genotype (WT vs. *spr2*) and treatment (mycorrhizal vs. non-mycorrhizal) on the expression of a battery of selected genes was calculated using the $2^{-\Delta Ct}$ method (*Livak & Schmittgen, 2001*). Transcript abundance data were normalized against the average transcript abundance of two reference genes: *TIP41* and *SAND* (*Expósito-Rodríguez et al., 2008*). Values reported are those of 3 independent mycorrhization experiments each one of which was analyzed using mRNA extracted from a single pooled sample prepared by combining the roots of all 10 plants used per experiment. Gene expression data represent the mean ± SE of three technical replicates per combined root pool per independent experiment. qPCR raw data are available from Zenodo (https://zenodo.org/), DOI 10.5281/zenodo.3560410.

## Statistical analysis of mycorrhizal colonization experiments

Data were analyzed by one-way ANOVA to determine whether or not the means of the different treatments tested were equal. A multiple comparison procedure with the Tukey's test was performed to find significant differences between means. All tests were conducted using the Minitab 15 statistical software package (Minitab Inc., State College, PA, USA). Differences at $p < 0.05$ were considered as statistically significant.

# RESULTS

## Mycorrhizal colonization is compromised in *spr2* mutants

Three independent experiments were performed. AMF colonization parameters, irrespective of the AMF used as inoculum and method employed to evaluate colonization, were significantly higher in roots of WT plants compared to *spr2* mutants. The AMF colonization parameters of two independent experiments sampled at 32 days after inoculation (dpi) (E3; Fig. 1A) and 45 dpi (E2; Fig. 1B), were determined using the magnified intersections method (*McGonigle et al., 1990*). They were very similar to those of a previous experiment, sampled at 50 dpi (E1), in which AMF colonization was determined using the MYCOCALC software (Fig. S1). Apart from the gene expression

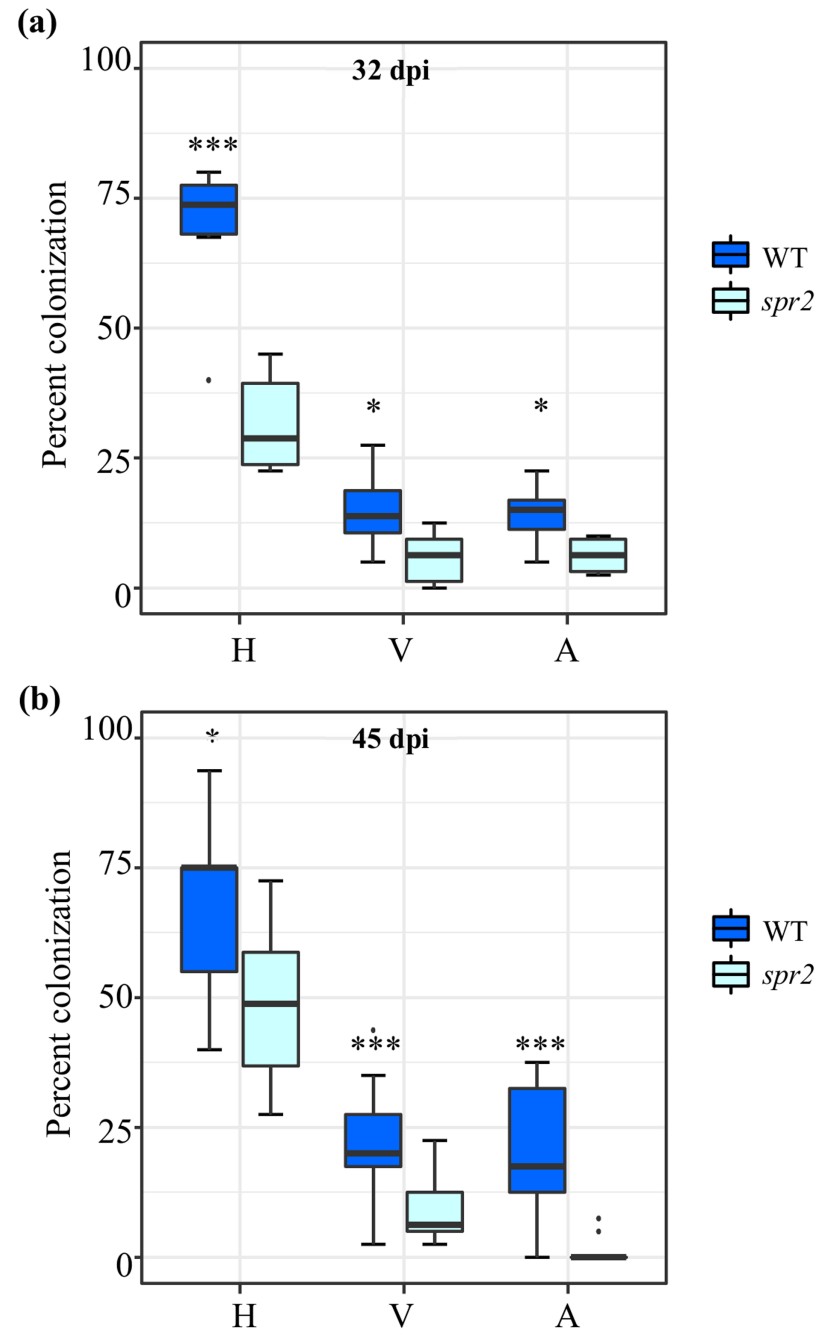

**Figure 1 Degree of arbuscular mycorrhizal fungi (AMF) colonization at (A) 32 days post inoculation (dpi) and (B) 45 dpi, in roots of wild type (WT) and *spr2* tomato plants inoculated with a consortium of six AMF species.** Box-and-whisker plots show high, low, and median percentage values of hyphae (H), vesicles (V) and arbuscules (A) quantified in in the root system of wild type (WT) and *spr2* mutant tomato plants. Asterisks over the box-and-whisker plots represent statistically significant differences at $p \leq 0.05$ (*) or 0.001 (***) (Tukey Kramer test; $n = 10$).   

analysis, all subsequent data was obtained from leaf and root tissues derived from the E2 and E3 experiments.

Similar to a previous characterization of the *spr2* tomato mutant (*Li et al., 2003*), no significant differences in the the growth rate and morphology of *spr2* and WT plants were observed during the experiments herewith described. However, measurement of physiological parameters during the course of E2, indicated that mycorrhizal colonization promoted plant growth, determined as plant height, in WT plants. No growth promotion was recorded in *spr2* plants (Table S2). Irrespective of the genotype, photosynthetic parameters were not altered by mycorrhizal colonization. However, contrary to reported data (*Zare-Maivan, Khanpour-Ardestani & Ghanati, 2017*), AMF colonization had a contrasting effect on chlorophyll content; the significant increase detected in leaves of mycorrhizal *spr2* plants was opposite to the tendency toward lower chlorophyll content observed in equivalent WT leaves (Table S2).

## Mycorrhizal-induced changes in jasmonic (JA) and salicylic acid (SA) content in leaf and roots

JA and SA levels were measured in leaves and roots of plants sampled at 45 dpi. At this sampling point, significantly increased JA contents were detected in both leaves and roots of mycorrhizal WT plants. The effect was more pronounced in leaves.
No increase in JA content was produced in equivalent organs of mycorrhizal *spr2* plants (Figs. 2A and 2B). Mycorrhizal colonization had no effect on SA levels in roots and leaves of WT. SA content in leaves of *spr2* plants was lower than WT plants and was not modified by AMF colonization, while a significant increase in SA levels was detected in mycorrhizal *spr2* roots (Figs. 2C and 2D). No SA-glycosides were detected.

## Gene expression profiles

In WT and *spr2* roots, AMF colonization led to several changes in the expression levels of a selected set of genes, which were analyzed at 32, 45 and 50 dpi (Table 1). In general, changes in gene expression were more intense and frequent in roots analyzed at 32 dpi and tended to decrease as the mycorrhizal colonization period extended from 45 to 50 dpi. The gene expression profiles obtained were the following:

### Phosphate transporter LePT4

*LePT4* is considered a reliable indicator of mycorrhizal-colonization in roots of *Medicago truncatula* and rice (*Javot et al., 2007*; *Wang et al., 2017*). In tomato, AMF colonization also induced the accumulation of *LePT4* transcripts. Unexpectedly, *LePT4* expression was higher in mycorrhizal *spr2* roots sampled at 32 and 45 dpi having significantly reduced colonization levels. Irrespective of the genotype, *LePT4* transcript abundance decreased ca. 3-fold in mycorrhizal roots sampled at 45 dpi. This pattern was reversed at 50 dpi, time at which the *LePT4* expression levels coincided with the degree of AMF colonization intensity.

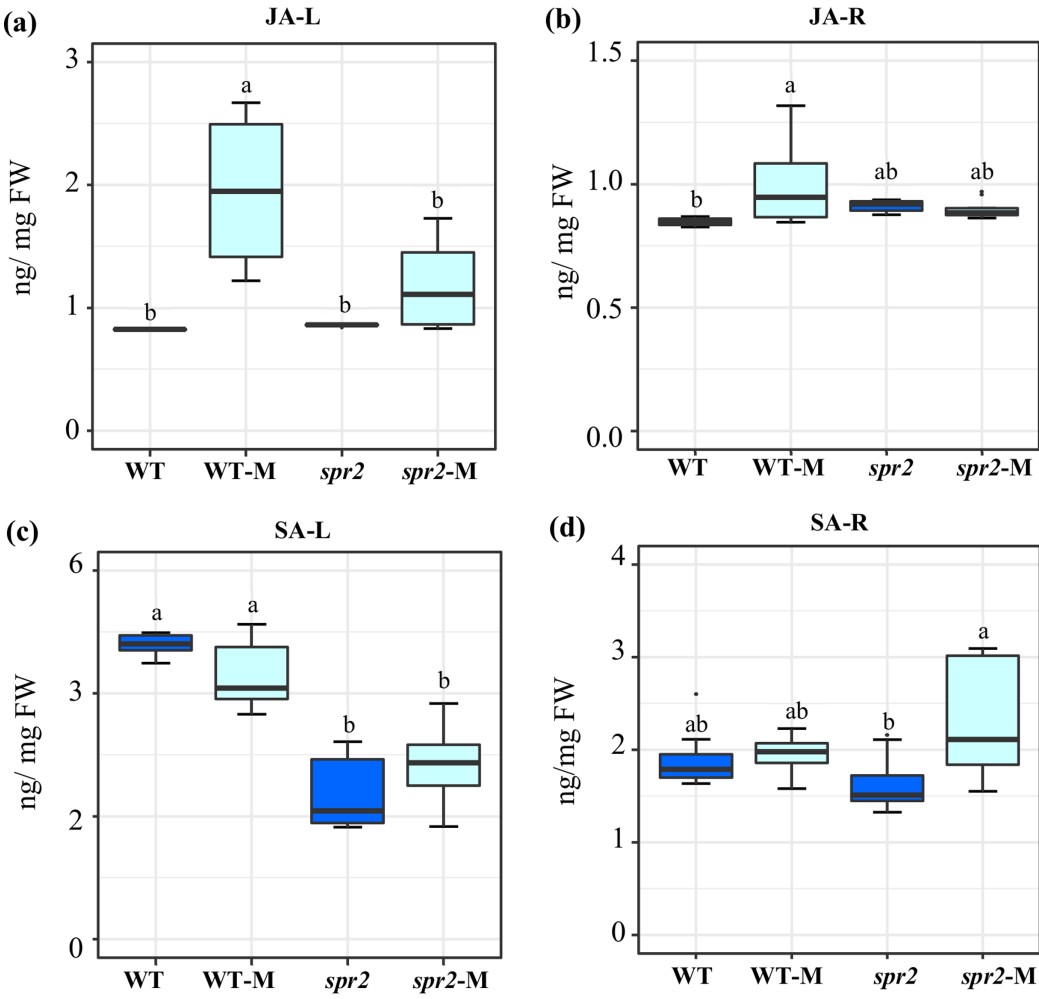

**Figure 2 Modification of jasmonic acid (JA) and salicylic acid (SA) levels in response to mycorrhizal colonization.** Content of (A) and (B) JA, and (C) and (D) SA were determined in leaves (JA-L, SA-L) and roots (JA-R, SA-R) of control wild-type (WT) and *spr2* mutant tomato plants or of mycorrhizal WT (WT-M) and *spr2* (*spr2*-M) plants sampled 45 dpi. Box-and-whisker plots show high, low, and median values. Different letters over the box-and-whisker plots represent statistically significant differences at $p \leq$ 0.05 (Tukey Kramer test; $n = 10$).

### Phytohormone-related genes

The expression of three genes involved in brassinosteroid (BR) biosynthesis was mostly repressed in mycorrhizal WT roots, contrary to the widespread induction observed in colonized *spr2* roots. Except for slight variations, the expression pattern was similar in roots harvested at 32, 45 and 50 dpi. The genes in question were *DE-ETIOLATED 2* (*DET2*) coding for a steroid 5a-reductase active in the early steps of the BR biosynthetic pathway (*Wei & Li, 2016*), *DWARF4* (*DWF4*) and *CONSTITUTIVE PHOTOMORPHOGENESIS AND DWARFISM* (*CPD*), both coding for P450 enzymes involved in downstream BR biosynthesis steps (*Wei & Li, 2016*).

The *ENT-COPALYL DIPHOSPHATE SYNTHASE* (*CPS*) and *ENT-KAURENE SYNTHASE* (*KS*) genes code for enzymes catalyzing the initial steps in the gibberellin (GA)

**Table 1 Gene expression levels in roots of mycorrhizal wild-type (WT) and *spr2* mutant tomato plants.** The expression levels[1] shown were obtained from non-mycorrhizal roots and mycorrhizal roots colonized for 32, 45 and 50 dpi.

| Colonization period | 32 dpi | | | | | | 45 dpi | | | | | | 50 dpi | | | | | |
|---|---|---|---|---|---|---|---|---|---|---|---|---|---|---|---|---|---|---|
| Genotype/Treatment | WT (C)[2] | WT (M)[3] | WT (M/C)[4] | spr2 (C) | spr2 (M) | spr2 (M/C) | WT (C) | WT (M) | WT (M/C) | spr2 (C) | spr2 (M) | spr2 (M/C) | WT (C) | WT (M) | WT (M/C) | spr2 (C) | spr2 (M) | spr2 (M/C) |
| **Gene/Gene category** | | | | | | | | | | | | | | | | | | |
| *Mycorrhizal colonization marker* | | | | | | | | | | | | | | | | | | |
| LePT4 | 0.008 | 4.46 | **560**[6] | 0.003 | 4.76 | **1746** | 0.02 | 2.52 | **140** | 0.004 | 1.73 | **403** | 0.001 | 1.97 | **2377** | 0.003 | 0.50 | **185** |
| *Brassinosteroids* | | | | | | | | | | | | | | | | | | |
| DET 2 | 0.55 | 0.24 | **0.44**[7] | 0.27 | 0.66 | **2.45** | 0.70 | 0.29 | 0.42 | 0.28 | 0.30 | 1.09 | 1.15 | 2.16 | **1.88** | 0.81 | 0.69 | 0.85 |
| DWF4 | 0.27 | 0.14 | **0.53** | 0.07 | 0.27 | **3.77** | 0.10 | 0.02 | 0.22 | 0.05 | 0.10 | **2.03** | 0.31 | 0.10 | **0.32** | 0.24 | 0.25 | 1.04 |
| CPD | 0.34 | 0.15 | **0.45** | 0.16 | 0.32 | **2.01** | 0.15 | 0.08 | 0.55 | 0.12 | 0.03 | **0.23** | 0.16 | 0.17 | 1.08 | 0.10 | 0.17 | **1.79** |
| *Gibberellins* | | | | | | | | | | | | | | | | | | |
| CPS | 0.003 | 0.05 | **14.30** | 0.004 | 0.03 | **8.30** | 0.02 | 0.07 | **4.17** | 0.012 | 0.006 | **0.48** | 0.01 | 0.10 | **7.80** | 0.009 | 0.011 | 1.21 |
| KS | 0.47 | 0.17 | **0.37** | 0.30 | 0.33 | 1.07 | 0.27 | 0.24 | 0.90 | 0.36 | 0.28 | 0.78 | 0.69 | 0.39 | **0.56** | 0.43 | 0.52 | 1.23 |
| GA3ox | 0.01 | 0.33 | **23.6** | 0.02 | 1.09 | **69.0** | 0.03 | 0.39 | **14.1** | 0.03 | 0.45 | **15.4** | 0.02 | 0.68 | **29.9** | 0.01 | 0.26 | **32.7** |
| GAI | 1.63 | 0.53 | **0.33** | 1.14 | 6.17 | **5.39** | 0.65 | 0.37 | **0.57** | 2.91 | 2.18 | 0.75 | 48.6 | 13.9 | **0.29** | 14.9 | 25.5 | **1.71** |
| *Abscisic acid* | | | | | | | | | | | | | | | | | | |
| TAS14 | –[5] | – | – | – | – | – | – | – | – | – | – | – | 248 | 375 | **1.51** | 444 | 746 | **1.68** |
| NCED1 | – | – | – | – | – | – | – | – | – | – | – | – | 0.99 | 1.96 | **1.99** | 1.29 | 1.92 | **1.49** |
| *Apocarotenoids* | | | | | | | | | | | | | | | | | | |
| CCD7 | 0.01 | 0.12 | **10.9** | 0.01 | 0.24 | **33.6** | 0.04 | 0.27 | **6.81** | 0.03 | 0.42 | **16.7** | 0.14 | 0.79 | **5.51** | 0.09 | 0.19 | **2.10** |
| CCD1a | – | – | – | – | – | – | – | – | – | – | – | – | 1.34 | 0.94 | 0.70 | 0.94 | 0.98 | 1.04 |
| CCD1b | 1.44 | 0.85 | **0.59** | 1.04 | 2.15 | **2.06** | 1.13 | 0.60 | **0.53** | 0.74 | 1.25 | **1.68** | 2.63 | 1.02 | **0.39** | 3.47 | 3.35 | 0.97 |
| *9-LOX* | | | | | | | | | | | | | | | | | | |
| HPL | – | – | – | – | – | – | – | – | – | – | – | – | 2.19 | 5.98 | **2.73** | 1.66 | 3.64 | **2.19** |
| LOXA | – | – | – | – | – | – | – | – | – | – | – | – | 3.67 | 3.81 | 1.04 | 0.75 | 1.08 | **1.44** |
| DES | – | – | – | – | – | – | – | – | – | – | – | – | 3.05 | 9.47 | **3.10** | 3.34 | 5.73 | **1.72** |
| AOS 3 | – | – | – | – | – | – | – | – | – | – | – | – | 1.12 | 7.18 | **6.43** | 0.03 | 0.17 | **5.19** |
| *13-LOX* | | | | | | | | | | | | | | | | | | |
| LOXD | 3.08 | 4.39 | **1.43** | 1.65 | 2.33 | **1.41** | 0.26 | 0.30 | 1.16 | 0.28 | 0.42 | **1.47** | 14.8 | 7.60 | **0.51** | 14.4 | 10.4 | 0.72 |
| AOS 1 | – | – | – | – | – | – | – | – | – | – | – | – | 4.49 | 3.68 | 0.82 | 2.85 | 2.81 | 0.98 |
| JAZ 2 | 1.78 | 9.99 | **5.62** | 3.81 | 5.72 | **1.50** | 0.44 | 0.33 | 0.75 | 0.16 | 0.28 | **1.71** | 10.2 | 17.3 | **1.70** | 12.9 | 4.46 | **0.34** |
| JMT | 0.98 | 1.53 | **1.56** | 0.63 | 2.16 | **3.44** | 0.27 | 0.76 | **2.85** | 0.23 | 1.12 | **4.87** | 0.59 | 1.81 | **3.09** | 0.86 | 0.62 | 0.72 |
| *Wound response* | | | | | | | | | | | | | | | | | | |
| PS | 0.0009 | 0.0012 | 1.33 | 0.001 | 0.003 | **3.68** | 0.0017 | 0.0016 | 0.93 | 0.0013 | 0.0010 | 0.79 | 0.0023 | 0.0047 | **2.01** | 0.0028 | 0.0011 | **0.39** |
| RBOH1 | 0.30 | 0.07 | **0.24** | 0.09 | 0.23 | **2.63** | 0.33 | 0.14 | **0.42** | 0.22 | 0.16 | 0.73 | 0.26 | 0.13 | **0.51** | 0.23 | 0.13 | **0.55** |
| LHA1 | 3.11 | 1.11 | **0.36** | 1.51 | 2.24 | **1.48** | 1.35 | 1.19 | 0.88 | 1.25 | 0.82 | 0.65 | 2.40 | 2.35 | 0.98 | 1.88 | 2.74 | **1.46** |
| PLA2 | – | – | – | – | – | – | – | – | – | – | – | – | 0.57 | 0.49 | 0.85 | 0.54 | 0.63 | 1.18 |
| PINII | 0.41 | 0.74 | **1.79** | 0.45 | 3.13 | **6.95** | 0.013 | 0.005 | **0.40** | 0.0025 | 0.0015 | **0.62** | 0.68 | 0.20 | **0.30** | 0.11 | 0.05 | **0.41** |

(Continued)

| Colonization period | 32 dpi | | | | | | 45 dpi | | | | | | 50 dpi | | | | | |
|---|---|---|---|---|---|---|---|---|---|---|---|---|---|---|---|---|---|---|
| Genotype/ Treatment | WT (C)[2] | WT (M)[3] | WT (M/C)[4] | spr2 (C) | spr2 (M) | spr2 (M/C) | WT (C) | WT (M) | WT (M/C) | spr2 (C) | spr2 (M) | spr2 (M/C) | WT (C) | WT (M) | WT (M/C) | spr2 (C) | spr2 (M) | spr2 (M/C) |
| AroGP3 | 0.21 | 0.04 | **0.22** | 0.08 | 0.12 | **1.51** | 0.61 | 0.30 | **0.49** | 0.72 | 0.45 | **0.63** | 1.27 | 0.27 | **0.21** | 0.53 | 0.49 | 0.92 |
| PPO | – | – | – | – | – | – | – | – | – | – | – | – | 4.87 | 9.58 | **1.97** | 0.82 | 1.73 | **2.11** |
| CYP2 | 25.8 | 19.3 | 0.75 | 14.5 | 20.6 | **1.42** | 39.2 | 30.6 | 0.78 | 51.4 | 43.2 | 0.84 | 72.4 | 122 | **1.69** | 19.2 | 52.0 | **2.70** |
| SCP | 0.21 | 0.07 | **0.35** | 0.23 | 0.30 | **1.28** | 0.72 | 0.35 | **0.49** | 0.69 | 0.38 | **0.55** | 0.35 | 0.25 | 0.72 | 0.18 | 0.33 | **1.81** |
| **Ethylene biosynthesis** | | | | | | | | | | | | | | | | | | |
| ACS2 | 0.03 | 0.01 | **0.47** | 0.05 | 0.15 | **2.87** | 0.03 | 0.01 | **0.40** | 0.02 | 0.03 | **1.46** | 0.47 | 0.11 | **0.23** | 0.24 | 0.11 | **0.44** |
| ACS6 | – | – | – | – | – | – | – | – | – | – | – | – | 25.5 | 16.8 | **0.66** | 7.26 | 11.7 | **1.61** |
| ACO4 | 3.13 | 2.22 | 0.71 | 3.24 | 5.70 | **1.76** | 2.53 | 3.20 | 1.27 | 3.54 | 1.77 | **0.50** | 47.3 | 40.2 | 0.85 | 12.6 | 25.8 | **2.04** |
| **Ethylene signaling** | | | | | | | | | | | | | | | | | | |
| CTR4 | – | – | – | – | – | – | – | – | – | – | – | – | 2.21 | 3.35 | **1.52** | 1.94 | 1.64 | 0.85 |
| EIN2 | 1.00 | 0.30 | **0.30** | 0.23 | 0.34 | **1.49** | 1.12 | 0.73 | 0.65 | 1.07 | 0.12 | **0.11** | 8.32 | 6.49 | 0.78 | 4.70 | 14.07 | **2.99** |
| EIN3 | 0.27 | 0.17 | 0.64 | 0.20 | 0.58 | **2.86** | 0.25 | 0.25 | 0.99 | 0.21 | 0.09 | **0.45** | 1.45 | 1.29 | 0.89 | 1.50 | 5.80 | **3.86** |
| ERF1b | – | – | – | – | – | – | 0.84 | 0.82 | 0.98 | 1.12 | 0.42 | **0.38** | 0.20 | 2.40 | **12.03** | 0.09 | 0.97 | **11.15** |
| **Salicylic acid signaling** | | | | | | | | | | | | | | | | | | |
| PRI | 0.105 | 0.011 | **0.10** | 0.03 | 0.05 | **2.01** | 0.11 | 0.04 | **0.31** | 0.07 | 0.08 | 1.23 | – | – | – | – | – | – |
| SSI2 | 0.008 | 0.007 | 0.94 | 0.003 | 0.015 | **5.05** | 0.0014 | 0.0013 | 0.88 | 0.0023 | 0.0004 | **0.15** | – | – | – | – | – | – |
| SAMT | 0.006 | 0.002 | **0.34** | 0.01 | 0.04 | **3.81** | 0.14 | 0.10 | 0.68 | 0.31 | 0.42 | 1.35 | – | – | – | – | – | – |
| WRKY60 | 0.81 | 0.69 | 0.85 | 0.37 | 0.65 | **1.78** | – | – | – | – | – | – | – | – | – | – | – | – |
| **Phenylpropanoid/phenolic compounds biosynthesis** | | | | | | | | | | | | | | | | | | |
| PAL3 | 6.09 | 2.17 | **0.36** | 2.84 | 9.20 | **3.24** | 3.93 | 2.28 | **0.58** | 3.86 | 3.88 | 1.01 | 55.9 | 27.4 | **0.49** | 31.2 | 22.4 | 0.72 |
| PAL4 | 61.3 | 72.6 | 1.18 | 32 | 88 | **2.74** | 66.1 | 42.4 | 0.64 | 33.4 | 30.4 | 0.91 | 795 | 341 | **0.43** | 407 | 367 | 0.90 |
| PAL5 | – | – | – | – | – | – | – | – | – | – | – | – | 39.1 | 54.6 | **1.40** | 50.1 | 54.5 | 1.09 |
| FLS | – | – | – | – | – | – | – | – | – | – | – | – | 0.52 | 1.27 | **2.46** | 0.46 | 1.04 | **2.26** |
| **Isoprenoid biosynthesis/volatiles** | | | | | | | | | | | | | | | | | | |
| DXS-2 | – | – | – | – | – | – | – | – | – | – | – | – | 0.14 | 0.78 | **5.51** | 0.09 | 0.12 | 1.37 |
| FPS1 | 1.59 | 1.22 | 0.77 | 0.86 | 2.72 | **3.15** | 4.47 | 2.43 | **0.54** | 3.19 | 3.69 | 1.16 | 4.62 | 2.21 | **0.48** | 1.81 | 1.47 | 0.81 |
| HMGR1 | – | – | – | – | – | – | – | – | – | – | – | – | 4.56 | 2.94 | 0.64 | 4.34 | 4.53 | 1.04 |
| BEAT | – | – | – | – | – | – | – | – | – | – | – | – | 0.30 | 2.92 | **9.87** | 0.06 | 0.07 | 1.16 |
| **Glycoalkaloid biosynthesis** | | | | | | | | | | | | | | | | | | |
| GAME1 | 0.32 | 0.15 | **0.49** | 0.23 | 0.46 | **1.97** | 2.01 | 1.19 | **0.59** | 1.16 | 0.83 | 0.72 | 1.19 | 0.88 | 0.74 | 0.68 | 0.91 | 1.35 |
| CAS1 | 0.40 | 0.16 | **0.40** | 0.23 | 0.57 | **2.55** | 0.56 | 0.28 | **0.50** | 0.33 | 0.34 | 1.05 | 0.42 | 0.48 | 1.14 | 0.34 | 0.49 | **1.41** |

**Notes:**

[1] Gene expression levels were calculated using the $2^{-\Delta Ct}$ equation, according to *Livak & Schmittgen (2001)*.
[2] C, represents expression levels quantified in roots of non-mycorrhizal control plants.
[3] M, represents expression levels quantified in roots of mycorrhizal plants.
[4] M/C, represents the gene expression ratio between roots of mycorrhizal and non-mycorrhizal plants.
[5] The (–) symbol indicates that gene expression at these time points was not determined.
[6] M/C values in **bold text** indicate a positive influence on gene expression by AMF colonization.
[7] M/C values in **red bold text** indicate a negative influence on gene expression by AMF colonization.

biosynthetic pathway leading to the synthesis of *ent*-kaurene from geranyl to geranyl pyrophosphate in the chloroplast. The *GA 3-OXIDASE* (*GA3ox*) gene codes for an enzyme catalyzing the final steps in the formation of the biologically active GAs, consisting of a series of 3β-hydroxylations of GA-precursors (*Hedden & Thomas, 2012*). Except for *KS*, downregulated at 32 and 50 dpi in WT mycorrhizal roots, both *CPS* and *GA3ox* tended to be induced in AMF-colonized roots, irrespective of the genotype. On the other hand, the expression of the *GIBBERELLIC ACID INSENSITIVE* (*GAI*) gene, coding for a DELLA protein responsible for repressing GA-dependent signaling (*Peng et al., 1997*; *Schwechheimer, 2012*) showed contrasting patterns of expression, remaining repressed in mycorrhizal WT roots along all sampling points, whereas it was mostly induced in equivalent *spr2* roots.

The induced expression of the two ABA marker genes was similar in mycorrhizal WT and *spr2* roots sampled at 50 dpi. These included *9-CIS-EPOXYCAROTENOID DIOXYGENASE 1* (*NCED1*), coding for one of the two tomato NCED enzymes that catalyze a key rate-limiting step of ABA biosynthesis (*Martín-Rodríguez et al., 2016*) and *ABSCISIC ACID AND ENVIRONMENTAL STRESS-INDUCIBLE PROTEIN 14* (*TAS14*), coding for a tomato dehydrin induced by osmotic stress and ABA (*Godoy et al., 1994*).

### Apocarotenoid biosynthesis-related genes

The *CCD7* gene codes for one of the two dioxygenases required for strigolactone biosynthesis, an essential component of the initial plant-AMF communication (*López-Ráez et al., 2015*). Its expression was steadily induced at 32–50 dpi in both WT and *spr2* mycorrhizal roots, although expression levels tended to be higher in mycorrhizal *spr2* roots. The *CCD1b* gene, coding for a carotenoid dioxygenase enzyme involved not only in the biosynthesis of C13/C14 apocarotenoids, but also in the generation of apocarotenoid cleavage products, such as β-ionone, $C_{13}$ α-ionol and $C_{14}$ mycorradicin at late stages of the symbiosis (*López-Ráez et al., 2015*) was mostly induced in *spr2* roots, although at lower levels than *CCD7*. It's expression was widely repressed in mycorrhizal WT roots.

### 9-LOX and 13-LOX pathway genes

Several emblematic 9-LOX pathway genes were induced to similar levels in response to AMF colonization in mycorrhizal WT and *spr2* roots sampled at 50 dpi. Among the octadecanoid pathway genes leading to JA biosynthesis (*Ryan, 2000*) that were analyzed, *LIPOXYGENASE D* (*LOXD*) was induced to similar levels in WT and *spr2* mycorrhizal roots at 32 dpi. Its expression was subsequently reduced, particularly in WT mycorrhizal roots sampled at 50 dpi. Nomycorrhizal-induced changes between genotypes were detected in the expression of *ALLENE OXIDE SYNTHASE* (AOS). In contrast, the *JASMONIC ACID CARBOXYL METHYLTRANSFERASE* (*JMT*) and the *JASMONATE ZIM DOMAIN2* (*JAZ2*) genes involved in downstream JA signaling remained up-regulated in mycorrhizal WT roots during all three sampling points examined. JAZ2 is a member of the JAZ family of master regulators of the JA signaling pathway (*Chung et al., 2009*), while JMT is required for the synthesis of MeJA, an active JA volatile signal (*Seo et al., 2001*).

### Wound-response (WR) genes

Only a few marker genes of the JA-dependent wound response (*Ryan, 2000*) were induced in mycorrhizal WT roots. These were *PINII*, coding for an emblematic protein inhibitor that was up-regulated at 32 dpi. Others were *PS*, coding for the prosystemin precursor protein, and *PPO* and *CYP2*, two late genes coding for a polyphenol oxidase and a cysteine protease, respectively, that were induced at 50 dpi. All other WR genes were repressed in these roots in at least one sampling time point. The negatively affected genes were *AROGP3*, coding for the JA-regulated tomato polygalacturonase non-catalytic subunit gene (*Bergey et al., 1999*), *RESPIRATORY BURST OXIDASE HOMOLOG1* (*RBOH1*), an early WR gene proposed to contribute to the generation of ROS in wounded plants, *LHA1*, an early WR gene coding for a member of the numerous plasma membrane H$^+$-ATPase gene family in tomato (*Schaller & Oecking, 1999*; *Ryan, 2000*; *Liu et al., 2016*) and *SERINE CARBOXYPEPTIDASE* (*SCP*), another late WR marker gene (*Ryan, 2000*). Most of these genes, in addition to the late *CYSTEINE PROTEINASE* 2 (*CYP2*) WR marker gene, were induced in *spr2* mycorrhizal roots in at least one sampling time point, principally at 32 dpi.

### Ethylene (ET) biosynthesis- and signaling-related genes

The *CONSTITUTIVE TRIPLE-RESPONSE4* (*CTR4*) gene, coding for a tomato CTR-like protein that is homologous to an *Arabidopsis* Raf mitogen-activated protein triple kinase that suppresses ET signaling (*Adams-Phillips, Barry & Giovannoni, 2004*; *Wang et al., 2018b*) was induced in mycorrhizal WT roots sampled at 50 dpi. Most other ET-related genes were either unaffected or were repressed by AMF colonization in WT roots, that is, the ET biosynthetic and signaling *1-AMINOCYCLOPROPANE-1-CARBOXYLATE SYNTHASE2* (*ACS2*) and *ETHYLENE INSENSITIVE2* (*EIN2*) genes, respectively. This was contrary to the widespread induction of the ET biosynthetic genes *1-AMINOCYCLOPROPANE-1-CARBOXYLIC ACID (ACC) SYNTHASE6* (*ACS6*) and *ACC OXIDASE4* (*ACO4*) (*Wang, Li & Ecker, 2002*), and of genes coding for activators of ET signaling and response such as *EIN2* and *EIN3* (*Mata et al., 2018*) in mycorrhizal *spr2* roots. Activation/repression of these genes was recorded primarily in roots sampled at 32 dpi. The induction of the ET-related *ERF1b* transcription factor (TF) gene was recorded in mycorrhizal roots of both WT and *spr2* plants sampled at 50 dpi.

### Salicylic acid (SA) signaling-related genes

SA-related genes were analyzed considering previous findings reporting that the *FAD7* mutation in the *spr2* plants positively modulates SA signaling (*Avila et al., 2012*). Accordingly, the *PATHOGENESIS-RELATED GENE1* (*PR1*), a SA responsive gene induced in response to pathogens and associated with the hypersensitive response in tomato (*Tornero et al., 1997*) was induced in mycorrhizal *spr2* roots sampled at 32 dpi, but extensively repressed in mycorrhizal WT roots. However, the *SUPPRESSOR OF SA INSENSITIVITY2* (*SSI2*) gene encoding a stearoyl acyl carrier protein involved in the conversion of stearic acid (C18:0) into oleic acid (C18:1) was induced in mycorrhizal *spr2* roots at 32 dpi, but repressed at 45 dpi. The late repression of this gene could have

contributed to reduce oleic acid levels, and concomitantly to increase SA content, as reported previously in Arabidopsis, soybean and other plants (*Lim et al., 2017*). In addition, the *SA CARBOXYL METHYLTRANSFERASE1* (*SMT*) gene that leads to the formation of volatile methyl salicylate (MeSA), considered to act as a mobile signal for the systemic acquired resistance (SAR) response (*Park et al., 2007*) and a WRKY transcription factor gene associated with SA signaling (*Van Verk et al., 2008*; *Gallou, Declerck & Cranenbrouck, 2012*; *Li et al., 2019*), were induced in mycorrhizal *spr2* roots, generally at 32 dpi. The *SMT* gene was repressed in mycorrhizal WT roots sampled at 32 dpi.

### Secondary metabolism biosynthetic genes

The analysis of key phenylpropanoid genes indicated that *PAL3* and *PAL4* were widely repressed in mycorrhizal WT roots, whereas they were induced in colonized *spr2* roots sampled at 32 dpi. The expression of *PAL5* was induced by mycorrhizal colonization in WT roots sampled at 50 dpi. These multi-member gene families (*Chang et al., 2008*), code for enzymes catalyzing the rate-limiting step of the phenylpropanoid biosynthetic pathway responsible for the generation of a high diversity of phenolic substances, including flavonoids. The *FLAVONOL SYNTHASE* (*FLS*) gene coding for an enzyme catalyzing a key step in the synthesis of biologically active flavonols that presumably play a regulatory role in the mycorrhizal symbiosis (*Steinkellner et al., 2007*; *Mandal, Chakraborty & Dey, 2010*) was induced in both mycorrhizal WT and *spr2* roots at 50 dpi. The induction of the *1-DEOXY-D-XYLULOSE 5-PHOSPHATE2* (*DXS-2*) gene in mycorrhizal WT roots sampled at 50 dpi was positively related to higher AMF colonization. This was consistent with the fact that this gene codes for the critical regulatory enzyme of the plastidial methyl-erythritol-4-phosphate isoprenoid biosynthetic pathway leading to the generation, among many others, of C13–C14 apocarotenoid precursors. The *BENZYL ALCOHOL ACETYLTRANSFERASE* (*BEAT*) gene, coding for an enzyme known to participate in synthesis of scent volatiles via benzenoid metabolic pathways (*Bera, Mukherjee & Mitra, 2017*), was also induced only in mycorrhizal WT roots. In contrast, the *FARNESYL-DIPHOSPHATE SYNTHASE1* (*FPS1*) gene, coding for a branch point enzyme of the isoprenoid pathway leading to both sesquiterpenes and sterols, and also considered to be a distal regulatory point of 3-hydroxy-3-methylglutaryl-coenzyme A reductase (HMGR; *Szkopiṅska, 2000*), remained unchanged or was repressed at late colonization stages in mycorrhizal WT roots. It was, however, induced in equivalent *spr2* roots at 32 dpi. Two genes involved in the general pathway of steroid biosynthesis leading to phytosterols and the α-tomatine glycoalkaloid (*Itkin et al., 2011*; *Moses, Papadopoulou & Osbourn, 2014*; *Jin, Lee & Kim, 2017*) were tested. *CYCLOARTENOL SYNTHASE* (*CAS1*) was induced in AMF colonized *spr2* roots at 32 and 50 dpi but extensively repressed in equivalent WT roots. A similar expression pattern was shown by the *GLYCOALKALOID METABOLISM 1* (*GAME1*) gene.

### Targeted metabolic profile of tomato roots and leaves

The results obtained from a GC–MS analysis designed to detect selected metabolites, predominantly associated with primary metabolism in roots and leaves of control and

**Table 2 Effect of mycorrhizal colonization on metabolite abundance in roots of wild-type (WT) and *spr2* mutant tomato plants.** Metabolite quantitation was determined in roots of control and mycorrhizal plants of both genotypes, sampled after a 45 dpi mycorrhizal colonization period[1].

| Metabolites | Genotype/treatment ratios[2] | | | |
|---|---|---|---|---|
| | WT/ *spr2* | WT-M/WT | *spr2*-M/ *spr2* | WT-M/ *spr2*-M |
| 1,3-Pentadiene | 0.81[4] | 2.10[3] | 1.09 | 1.56 |
| Glyoxylic oxime acid | 0.76 | 1.61 | 0.72 | 1.72 |
| Phosphoric acid | 0.76 | 2.35 | 2.27 | 0.78 |
| n-Butylamine | 1.07 | 1.36 | 0.73 | 2.00 |
| Diethylene glycol | 0.85 | 2.29 | 1.34 | 1.46 |
| 4, 6-Dimethyl dodecane | 1.11 | 2.13 | 1.72 | 1.38 |
| Glycerol | 0.42 | 2.17 | 0.91 | 1.01 |
| Maleic acid | 0.66 | 1.30 | 0.82 | 1.05 |
| Succinic acid | 0.76 | 1.48 | 1.11 | 1.01 |
| Propanoic acid | 0.83 | 1.47 | 1.34 | 0.91 |
| Fumaric acid | 0.76 | 1.68 | 1.30 | 0.98 |
| Furanone | 0.90 | 2.11 | 1.69 | 1.13 |
| DL-malic acid | 0.81 | 0.95 | 1.09 | 0.71 |
| L-proline | 0.51 | 1.63 | 0.69 | 1.21 |
| m-Hydroxybenzoic acid | 1.04 | 1.60 | 2.30 | 0.72 |
| L-threonic acid | 0.56 | 1.32 | 1.16 | 0.64 |
| 1-Cyclohexene-3, 4,5-trihydroxy-1-carboxylic acid | 1.46 | 1.04 | 1.70 | 0.90 |
| Isocitric acid | 1.02 | 0.86 | 0.91 | 0.97 |
| D-fructose | 1.29 | 0.80 | 0.74 | 1.40 |
| D-glucose | 1.07 | 0.75 | 0.67 | 1.21 |
| Myo-inositol | 0.98 | 0.82 | 1.01 | 0.79 |
| D-glucuronic acid | 1.14 | 1.55 | 1.50 | 1.17 |
| Sucrose | 1.18 | 0.75 | 0.87 | 1.02 |

Notes:
[1] The targeted metabolite analysis was performed by GC–MS.
[2] The ratios were determined using the mean peak areas of each compound.
[3] Numbers in bold in a green background indicate a significantly higher ratio.
[4] Numbers in bold in a orange background indicate a significantly lower ratio.

mycorrhizal tomato plants analyzed at 45 dpi is shown in Tables 2 and 3. Different tendencies in metabolite variation patterns were detected in roots (Table 2). A comparison of the roots of non-mycorrhizal WT and *spr2* plants, showed that metabolite abundance was predominantly higher in roots of *spr2* plants, since 10 of 14 metabolites affected in the mutant were significantly increased in *spr2* roots. Relevant exceptions were glucose, fructose and sucrose. AMF colonization had a positive effect on root metabolite accumulation, considering that in both WT and *spr2* mycorrhizal roots, 10 of 12 and 13 significantly impacted metabolites, respectively, were increased in response to AMF colonization. Isocitric was significantly reduced in mycorrhizal roots of both genotypes, while proline ratio was significantly increased in mycorrhizal WT roots but reduced in equivalent *spr2* roots. Those that increased their abundance in mycorrhizal roots of both WT and *spr2* roots were $PO_4^{3-}$, diethylene glycol, succinic, propanoic, and fumaric organic

**Table 3 Effect of mycorrhizal colonization on metabolite abundance in leaves of wild-type (WT) and *spr2* mutant tomato plants.** Metabolite quantitation was determined in leaves of control and mycorrhizal plants of both genotypes, sampled after a 45 dpi mycorrhizal colonization period[1].

| Metabolites | Genotype/treatment ratios[2] | | | |
|---|---|---|---|---|
| | WT/ *spr2* | WT-M/WT | *spr2*-M/ *spr2* | WT-M/ *spr2*-M |
| 1,3-Pentadiene | 1.34 | 0.80 | 0.92 | 1.17 |
| Glyoxylic oxime acid | 1.09 | 1.10 | 0.88 | 1.36 |
| Propanoic acid | 1.25 | 0.87 | 0.88 | 1.24 |
| Phosphoric acid | **1.53**[3] | 1.38 | **0.72** | **2.92** |
| 2-Butenoic acid | **5.36** | 0.72 | 1.29 | **3.00** |
| Benzoic acid | 1.50 | 0.93 | 0.75 | **1.87** |
| Diethylene glycol | 1.13 | 1.01 | **0.78** | **1.48** |
| Serine | **0.22**[4] | **3.64** | **0.49** | 1.63 |
| 4,6-dimethyl dodecane | 1.06 | 1.26 | 0.69 | **1.92** |
| Glycerol | **0.77** | 0.99 | 1.11 | **0.69** |
| Maleic acid | **0.53** | 1.09 | **1.33** | **0.43** |
| Succinic acid | **1.15** | 1.05 | 1.26 | 0.96 |
| Butanedioic acid, methyl | **1.19** | 1.00 | **0.68** | **1.76** |
| Propanoic acid | 1.10 | 1.09 | 0.83 | **1.44** |
| Fumaric acid | **0.85** | 1.07 | 0.93 | 0.98 |
| Furanone | 0.77 | **1.39** | 0.87 | 1.24 |
| L-aspartic acid | 1.37 | **0.55** | 1.27 | 0.59 |
| D-(-)-citramalic acid | 1.11 | 1.16 | 1.16 | 1.10 |
| DL-malic acid | 0.90 | 1.38 | 0.94 | **1.32** |
| L-proline, 5-oxo | **0.47** | **1.63** | 1.71 | **0.45** |
| m-Hydroxybenzoic acid | 1.02 | 1.05 | **0.68** | 1.57 |
| L-threonic acid | **1.72** | 0.78 | 1.09 | 1.23 |
| Phenylpyruvic acid | **1.68** | 0.77 | **0.82** | 1.57 |
| 1-Cyclohexene-3, 4, 5-trihydroxy-1-carboxylic acid | **2.19** | **0.61** | 0.90 | **1.48** |
| Isocitric acid | 1.07 | 0.89 | 1.00 | 0.96 |
| D-fructose | **0.62** | **1.44** | **1.20** | **0.74** |
| D-glucose | **0.54** | **1.40** | 1.03 | **0.74** |
| Myo-inositol | 1.11 | **0.84** | 0.92 | 1.01 |
| Phytol | 1.26 | 0.84 | 0.91 | 1.18 |
| D-glucuronic acid | **2.46** | **0.62** | **1.19** | **1.28f** |
| Sucrose | **1.22** | **0.87** | **0.80** | **1.33** |

Notes:
[1] The targeted metabolite analysis was performed by GC–MS.
[2] The ratios were determined using the mean peak areas of each compound.
[3] Numbers in bold in a green background indicate a significantly higher ratio.
[4] Numbers in bold in a orange background indicate a significantly lower ratio.

acids, furanone and L-threonic acid. Differences were minimal between mycorrhizal WT and *spr2* roots. Only the abundance of DL-malic acid and L-threonic acid was found to be significantly different between them, both being higher in AMF-colonized *spr2* roots.

Conversely, the abundance of 16/31 metabolites was significantly affected in leaves of *spr2* mutant plants compared WT plants (Table 3). Nine of these were decreased and

**Table 4 Effect of mycorrhizal colonization on fatty acid abundance in roots of wild-type (WT) and *spr2* mutant tomato plants.** Fatty acids were quantified in roots of control and mycorrhizal plants of both genotypes, sampled after a 45 dpi mycorrhizal colonization period[1].

| Fatty acid | % Abundance[2] | | Genotype/treatment ratios | | | |
|---|---|---|---|---|---|---|
| | WT | *spr2* | WT/ *spr2* | WT-M/WT | *spr2*-M/ *spr2* | WT-M/ *spr2*-M |
| C14:0 (myristic acid) | 0.40 | 0.28 | **1.42**[3] | 0.98 | 1.00 | **1.39** |
| C14:0 (13-methyl) | 0.69 | 0.53 | **1.32** | 0.92 | 0.98 | **1.23** |
| C15:0 | 0.62 | 0.48 | **1.28** | 0.99 | 1.02 | **1.25** |
| C15:0 (14-methyl) | 0.48 | 0.37 | **1.31** | 1.04 | 1.00 | **1.36** |
| C16:0 (palmitic acid) | 21.22 | 21.70 | 0.98 | **1.23** | 0.97 | **1.23** |
| C16:0 (15-methyl) | 2.92 | 2.39 | **1.22** | **0.93** | 1.03 | **1.10** |
| C16:1 (9Z) (palmitoleic acid) | 1.13 | 0.69 | **1.64** | 0.85 | 1.12 | 1.26 |
| C18:0 (stearic acid) | 3.24 | 2.38 | **1.36** | 0.93 | 1.09 | **1.17** |
| C18:1 (9Z) (oleic acid) | 2.33 | 1.33 | 1.75 | 1.17 | **1.09** | 1.86 |
| C18:1 (9E) (elaidic acid) | 0.86 | 1.26 | **0.69**[4] | **2.38** | 0.76 | **2.14** |
| C18:2 (linoleic acid) | 41.55 | 52.44 | **0.79** | 0.99 | 1.00 | **0.78** |
| C18:3 (linolenic acid) | 6.60 | 3.14 | **2.10** | 0.90 | 1.00 | **1.90** |
| C20:0 (eicosanoic acid) | 2.04 | 1.51 | **1.35** | **0.71** | 1.07 | 0.90 |
| C22:0 (behenic acid) | 10.69 | 7.94 | **1.35** | **0.70** | 1.01 | 0.94 |
| C24:0 (lignocceric acid) | 5.23 | 3.56 | **1.47** | **0.83** | 1.02 | **1.20** |

Notes:
[1] FA analysis was performed by GC–MS.
[2] % Abundance determined using the mean peak areas of each FA.
[3] Numbers in bold in a green background indicate a significantly higher ratio.
[4] Numbers in bold in a orange background indicate a significantly lower ratio.

7 were increased. Relevant differences were the reduction of 2-butenoic acid, D-glucuronic acid, 1-cyclohexene-3, 4, 5-1-carboxylic acid, L-threonic acid, $PO_4^{3-}$ and sucrose, while serine, oxoproline and maleic acid increased. Fructose and glucose levels were also higher in *spr2* leaves. Compared to non-mycorrhizal controls, AMF colonization modified the accumulation of 10 foliar metabolites in *spr2* mutant plants, predominantly in a negative way, since only 3 metabolites (i.e., maleic acid, fructose and D-glucuronic acid) showed significantly increased abundance. A comparison of leaf metabolite abundance between mycorrhizal WT and *spr2* plants followed this tendency, since only 5 of the 18 metabolites whose abundance significantly differed in response to AMF colonization in these plants, were found to increase in mycorrhizal *spr2* leaves. Apart from fructose, glucose, maleic acid and oxoproline, already mentioned above, glycerol was also among the metabolites that significantly increased their abundance in mycorrhizal *spr2* leaves.

## Fatty acids (FAs) in roots and leaves

The C16:2 $\Delta^{7E, 10E}$, hexadecatrienoic acid and parinaric FAs were not detected in roots of both genotypes, whereas compared to leaves of *spr2* plants (see below), C18:3 levels were only reduced 2.1-fold in *spr2* roots, while C18:2 levels were only ca. 1.3 higher (Table 4). In WT roots, AMF colonization increased the content of palmitic and elaidic FAs, while iso-methyl C16:0 (15 Me), eicosanoic (20:0), behenic (C22:0) and lignoceric (C24:0) FAs were significantly reduced. Oleic acid was the only FA whose abundance was

**Table 5 Effect of mycorrhizal colonization on fatty acid abundance in leaves of wild-type (WT) and *spr2* mutant tomato plants.** Fatty acids were quantified in leaves of control and mycorrhizal plants of both genotypes, sampled after a 45 dpi mycorrhizal colonization period[1].

| Fatty acids | % Abundance[2] | | Genotype/treatment ratios | | | |
|---|---|---|---|---|---|---|
| | WT | *spr2* | WT/ *spr2* | WT-M/WT | *spr2*-M/ *spr2* | WT-M/ *spr2*-M |
| C14:0 (myristic acid) | 0.08 | 0.06 | **1.39**[5] | 1.00 | 1.07 | **1.31** |
| C15:0 | 0.04 | 0.05 | 0.90 | 1.05 | 0.99 | 0.96 |
| C16:0 (palmitic acid) | 16.95 | 15.82 | **1.07** | **1.18** | 1.01 | **1.25** |
| C16:0 (14 Me) | 0.25 | 0.22 | **1.14** | **1.16** | 0.65 | **2.02** |
| C16:1 (7E) (palmitelaidic acid) | 1.06 | 1.46 | **0.72** | **1.30** | **0.77** | **1.23** |
| C16:1 (9Z) (palmitoleic acid) | 0.29 | 0.32 | **0.90** | 1.03 | 0.90 | 1.03 |
| C16:2 (palmitlinoleic acid) | 0.40 | 4.84 | **0.08**[6] | 1.02 | 1.06 | **0.08** |
| C16:3 | 6.58 | –[3] | –[4] | **0.85** | – | – |
| C18:0 (stearic acid) | 1.45 | 1.12 | **1.29** | **1.14** | **1.32** | **1.11** |
| C18:1 (9E) | 0.44 | 0.39 | 1.13 | 1.10 | 0.94 | **1.31** |
| C18:2 (linoleic acid) | 12.54 | 71.71 | **0.17** | **1.42** | 1.01 | **0.25** |
| C18:3 (linolenic acid) | 58.84 | 3.10 | **18.99** | **0.86** | 0.76 | **21.45** |
| C18:4 (parinaric acid) | 0.11 | – | – | **2.34** | – | – |
| C20:0 (eicosanoic acid) | 0.36 | 0.32 | 1.11 | **1.14** | 0.81 | **1.57** |
| C22:0 (behenic acid) | 0.27 | 0.31 | 0.89 | 1.12 | 1.03 | 0.97 |
| C24:0 (lignoceric acid) | 0.34 | 0.28 | **1.21** | 1.12 | 0.91 | **1.48** |

Notes:
[1] FA analysis was performed by GC–MS.
[2] % Abundance determined using the mean peak areas of each FA.
[3] Fatty acid not detected.
[4] Ratio could not be determined.
[5] Numbers in bold in a green background indicate a significantly higher ratio.
[6] Numbers in bold in a orange background indicate a significantly lower ratio.

increased in response AMF colonization in *spr2* roots. A comparison between mycorrhizal WT and *spr2* roots showed that FA content was, in general, significantly higher in WT roots, since 10/11 FAs, excepting linoleic acid, were significantly higher in mycorrhizal WT roots.

In leaves of WT plants, AMF colonization positively affected the amount of palmitic, the iso-methyl branched C16:0 (14 Me) FA, C16:1 ($\Delta^{7E}$), linoleic, parinaric (C18:4 $\Delta^{9Z, 11E, 13Z,15E}$), stearic and eicosanoic (C20:0) FAs, but decreased the content of cis-7, 10, 13 hexadecatrienoic acid (C16:3 $\Delta^{7E, 10E, 13E}$) and linolenic acid (Table 5). FA composition in *spr2* plants, most predominantly in leaves, accurately reflected the lost functionality of FATTY ACID DESATURASE7, as their levels of polyunsaturated FAs (i.e., C16:3, linolenic and parinaric acids), compared to WT leaves, were considerably reduced (e.g., an approximately 20-fold reduction in linolenic acid) or remained undetected (e.g., 16:3 and parinaric FAs), whereas the content of C16:2 $\Delta^{7E, 10E}$ and linoleic FAswas between 6- and 12-fold higher. However, only the content of stearic (increased) and palmitelaidic (decreased) FAs was significantly modified by AMF colonization in leaves of *spr2* plants (Table 5). A comparison between WT and *spr2* mycorrhizal leaves indicated that most FAs detected, except C16:2 $\Delta^{7E, 10E}$ and linoleic acid, were significantly more abundant in the former.

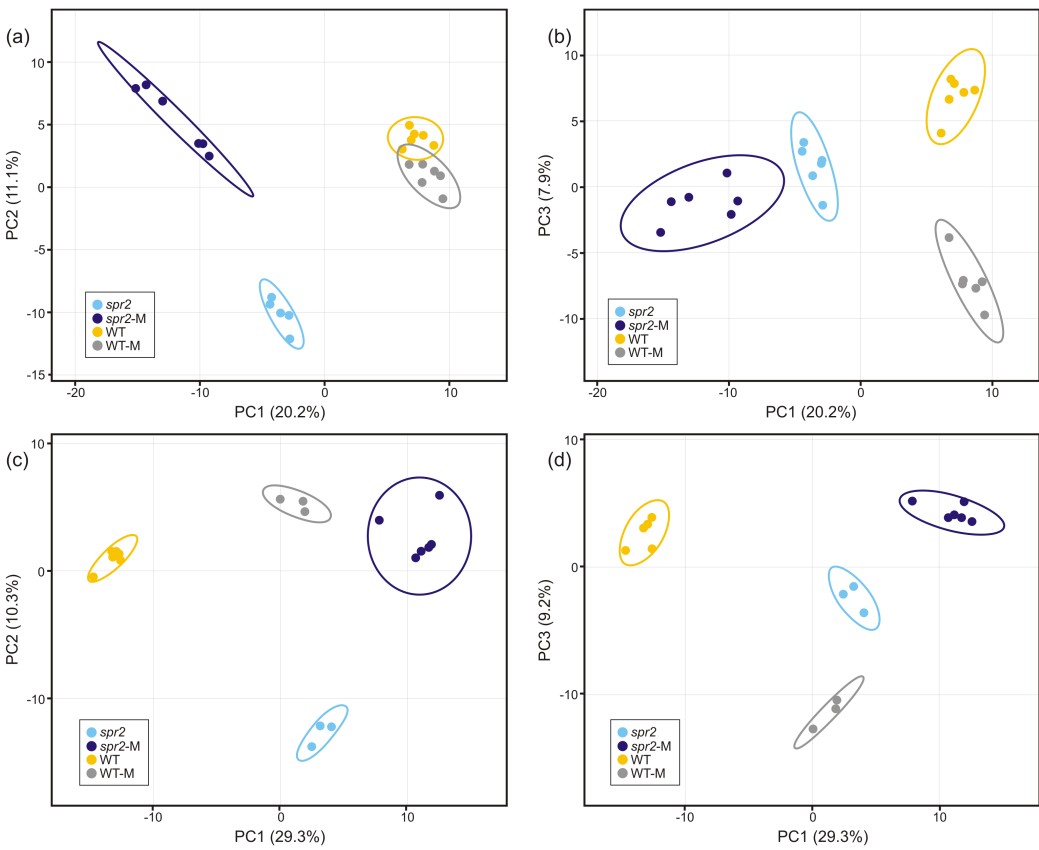

**Figure 3 Untargeted principal components analysis on metabolic fingerprinting of non-colonized and mycorrhizal wild-type (WT) and *spr2* mutant tomato roots.** The intensity of 367 metabolites sampled from WT and *spr2* roots at (A) and (B) 32 dpi, and (C) and (D) 45 dpi, was used to construct a matrix to evaluate the global variance at the metabolic level in WT and mutant *spr2* genotypes in response to AMF colonization (WT-M and *spr2*-M, respectively). The three principal components explain 39.2% and 48.8% of the total variance at 32 and 45 dpi, respectively. They were sufficient to separate the data by plant genotype and treatment. Prediction ellipses are such that they predict with 95% probability that any new observation from the same group will fall inside the ellipse ($n = 12$).

## Metabolic fingerprinting of tomato roots

Direct liquid introduction electrospray mass spectrometry (DLI–ESI–MS) fingerprints were generated to examine the effect of mycorrhizal colonization on the global metabolic profile in roots of WT and *spr2* mutant plants. PCA analysis of data from 367 significant metabolites identified in roots sampled at 32 and 45 dpi showed that at 32 dpi, the three PCs, representing 39.2% of the variance permitted the separation of root metabolomes by both treatment and genotype factors (Figs. 3A and 3B). Similar results were obtained with roots sampled from the 45 dpi experiment, where the separation of root metabolomes by treatment and genotype was possible with the three PCs representing 48.8% of the variance (Figs. 3C and 3D). A clear discrimination between genotype (WT, *spr2*), treatment (± AMF) and duration of treatment (32 vs. 45 dpi) was also revealed by the formation of separate clusters in the heat-map generated after HCA using the 100 most intense ions (Fig. 4).

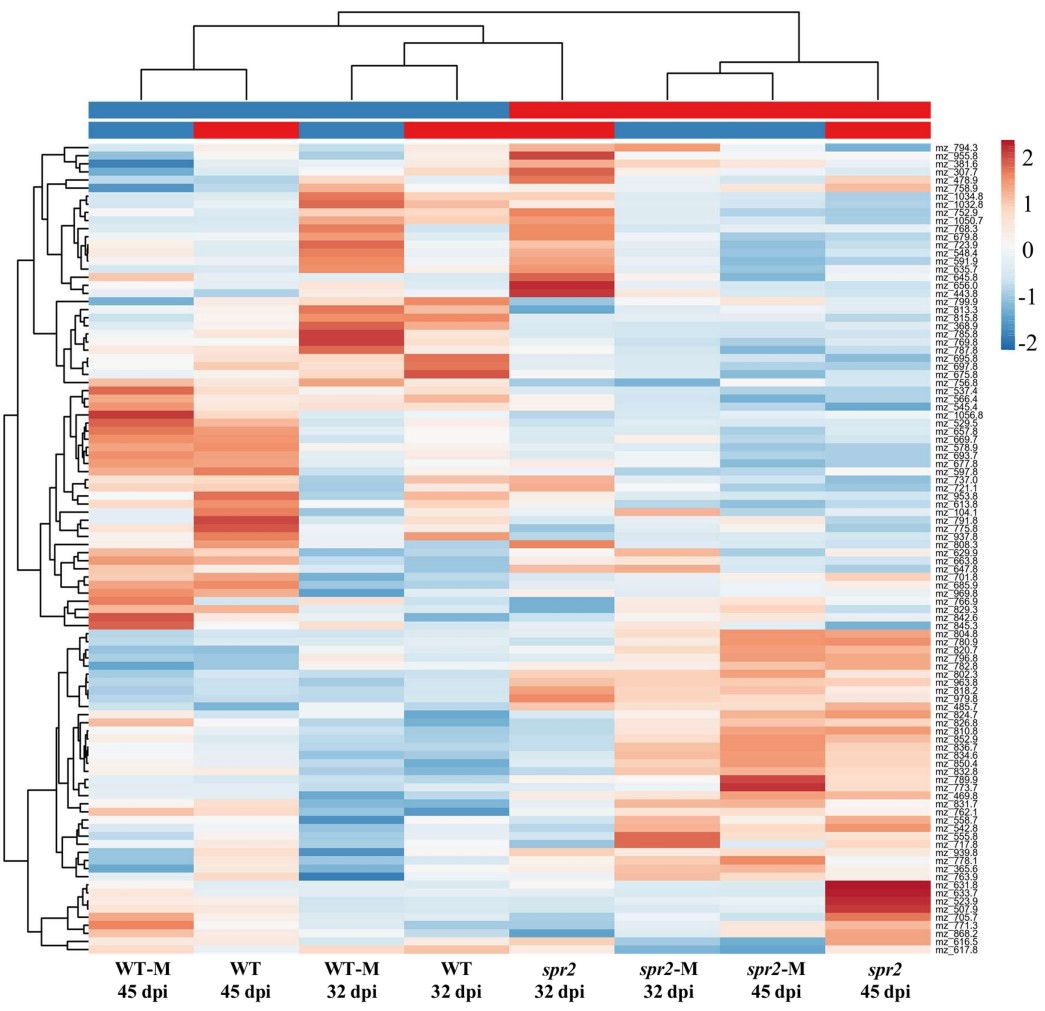

**Figure 4 Metabolic heat-map generated with the 100 most intense ions detected in roots of control wild-type (WT) and *spr2* mutant tomato plants or in roots of mycorrhizal WT (WT-M) and *spr2* (*spr2*-M) plants sampled at 32 and 45 dpi.** Ions in aqueous methanol root extracts were detected by DLI-ESI-MS as described in "Materials and Methods". The hierarchical clustering analysis resulted in a correct assignment of plants into well-defined clusters denoting genotype (WT vs. *spr2*), treatment (control vs. M) and treatment duration (32 vs. 45 dpi). The m/z values were extracted for putative assignation (refer to Tables S2 and S3).

Mass fingerprints were determined for each individual experiment. The results, shown in Tables S3 and S4, represent the putative metabolite ions whose significant change in abundance was associated with higher/lower AMF colonization levels at 32 and 45 dpi. They reinforced the above findings showing that the effect of AMF colonization on root primary and secondary metabolism was dependent on factors such as genotype and colonization time. They also agreed partially with the concept that AMF colonization efficiency involves changes in the content of several categories of lipids (see above), including lysophospholipids, in addition to phytosteroids (see below), carotenes, phenolic compounds, polyamines, auxins, cytokinins, amino acids and other nitrogen-containing compounds (*Akiyama & Hayashi, 2006*; *Drissner et al., 2007*; *Floss et al., 2008b*;

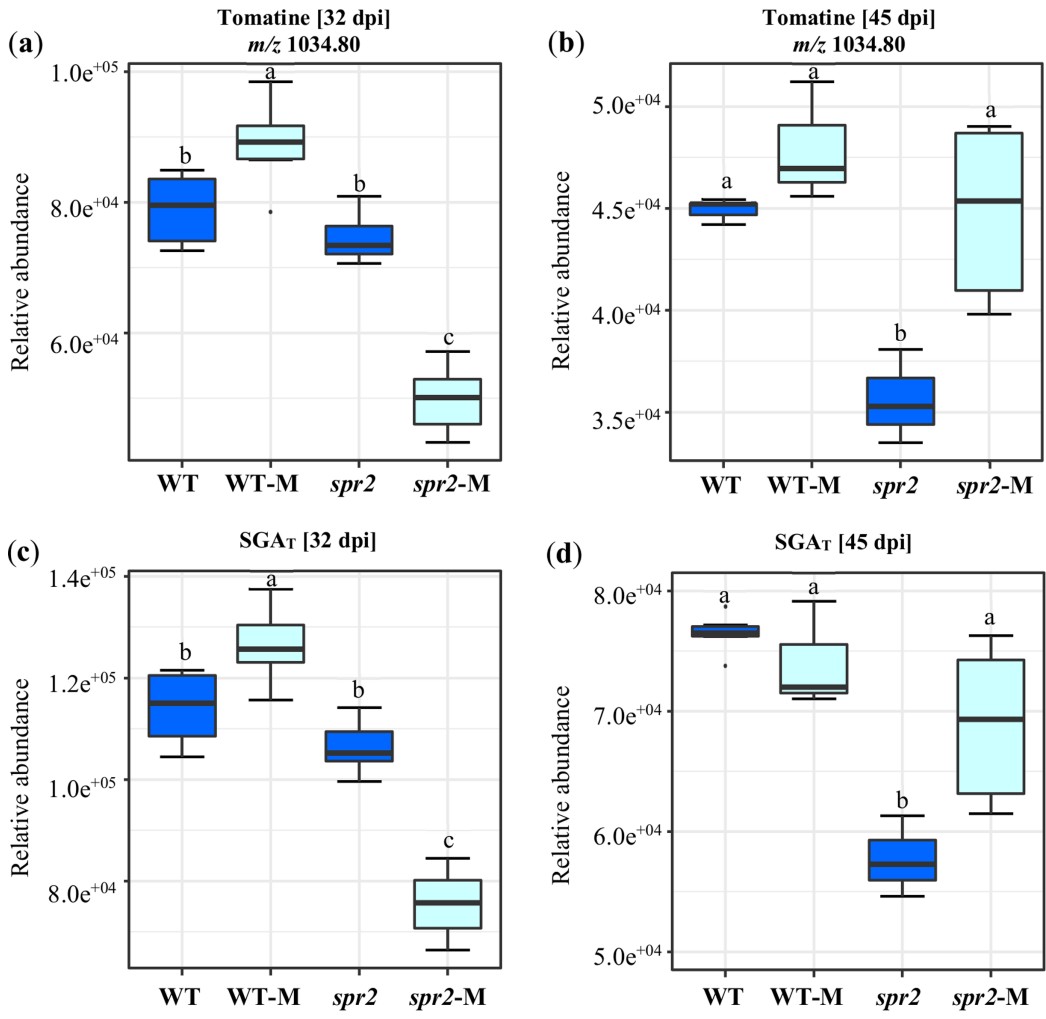

**Figure 5 Modification of tomatine and total steroid glycoalkaloid contents in response to mycorrhizal colonization.** Content of (A) and (B) tomatine, and (C) and (D) total steroid glycoalkaloid ions (SGA$_T$) determined in roots of control wild-type (WT) and *spr2* mutant tomato plants or of mycorrhizal WT (WT-M) and *spr2* (*spr2*-M) plants sampled at 32 and 45 dpi, respectively. Box-and-whisker plots show high, low, and median values. Different letters over the box-and-whisker plots represent statistically significant differences at $p \leq 0.05$ (Tukey Kramer test; $n = 10$).

*Floss & Walter, 2009*; *Whiteside, Garcia & Treseder, 2012*; *Jiménez-Bremont et al., 2014*; *Rivero et al., 2015*; *Bedini et al., 2018*; *Sánchez-Bel et al., 2018*; *Liao et al., 2018*).

A targeted assay focusing on the analysis of the tomatine biosynthetic pathway (*Montero-Vargas et al., 2018*) was performed. Differences were observed between genotypes and between non-colonized and colonized plants. They were also influenced by the duration of colonization. In general, the content of α-tomatine, its biosynthetic precursors and related catabolites (Fig. 5; Tables 6 and 7), was significantly lower in *spr2* mutant roots. This effect was more pronounced in roots sampled at 45 dpi. The influence of mycorrhizal colonization was also time-dependent in WT and *spr2* roots. At 32 dpi, significantly affected tomatine-related metabolites and total SGAs had a lower abundance in colonized *spr2* roots,

**Table 6 Changes in the abundance of α-tomatine and related compounds in roots at 32 dpi.** The content of α-tomatine, its biosynthetic precursors and catabolic products were determined in roots of wild-type (WT) and *spr2* mutant tomato plants after a 32 dpi mycorrhizal colonization period[1].

| Ion identity | Putative metabolite[1] | Wt/ *spr2* | Wt-M/Wt | *spr2*-M/ *spr2* | Wt-M/ *spr2*-M |
|---|---|---|---|---|---|
| mz_413.94 | Dehydrotomatine (tomatidenol) | 0.74 | 0.97 | 0.90 | 0.79 |
| mz_416.89 | Tomatidine and isomers | 0.91 | 0.97 | 1.09 | 0.81 |
| mz_576.89 | ND[2] | 0.95 | 1.13 | 0.98 | 1.10 |
| mz_578.93 | ND | 1.08 | 1.02 | 0.96 | 1.15 |
| mz_916.75 | Hydroxydehydrotomatidine Trihexoside and isomers | 0.79 | 1.19 | 0.76 | 1.23 |
| mz_1004.64 | Tomatidine dihexoside dipentoside and isomers | 0.97 | 1.16 | 0.71 | **1.58** |
| mz_1018.01 | Tomatidine + dihexoside + pentose + deoxyhexose | 1.63 | 0.74 | 1.02 | 1.18 |
| mz_1030.6 | Di-dehydrotomatine | 0.83 | **1.44** | 0.76 | **1.56** |
| mz_1032.75 | Dehydrotomatine and isomers | **1.28**[5] | **1.12** | **0.78** | **1.82** |
| mz_1034.8 | α-Tomatine and isomers | 1.06 | **1.13** | **0.67** | **1.78** |
| mz_1044.78 | UGA 4[3] | 0.77 | 1.20 | 0.84 | 1.10 |
| mz_1046.84 | UGA 5 | **0.80**[6] | **1.28** | **0.71** | **1.45** |
| mz_1050.7 | Hydroxytomatine and isomers | 0.93 | 1.04 | **0.54** | **1.79** |
| mz_1056.78 | ND | **1.22** | 0.92 | 1.09 | 1.03 |
| mz_1064.77 | Tomatidine tetrahexoside | 1.18 | 1.16 | 0.80 | **1.71** |
| mz_1072.72 | ND | **1.31** | 1.02 | 1.02 | **1.31** |
| mz_1090.77 | Acetoxy-dehydrotomatine | 1.06 | 0.96 | 0.80 | 1.26 |
| mz_1092.67 | Acetoxytomatine | 1.09 | 0.88 | **0.60** | **1.59** |
| | Total SGAs[4] | 1.07 | **1.11** | **0.71** | **1.67** |

**Notes:**
[1] Putative metabolite according to *Itkin et al. (2011)* Plant Cell 23:4507–4525.
[2] Identity of ion not determined.
[3] Unknown glycoalkaloid.
[4] Total steroid glycolalcaloids.
[5] Numbers in bold in a green background indicate a significantly higher ratio.
[6] Numbers in bold in a orange background indicate a significantly lower ratio.

while this pattern was reversed at 45 dpi (Fig. 5; Tables 6 and 7). Thus, α-tomatine content, which was significantly reduced in *spr2* roots in response to AMF colonization at 32 dpi, became enhanced at 45 dpi. In mycorrhizal WT roots, α-tomatine and most other SGAs tested remained significantly higher than controls only at 32 dpi (Figs. 5A and 5B; Tables 6 and 7).

# DISCUSSION

## Discernable biochemical and transcriptional changes occur in *spr2* tomato mutants compromised in mycorrhizal colonization in tomato

Tomato *spr2* mutant plants consistently showed reduced AMF colonization (Fig. 1; Fig. S1) and consequently were unable to benefit from the growth promotion effect usually produced by the AM symbiosis, as observed in mycorrhizal WT plants (Table S2). Low mycorrhization efficiency in *spr2* plants was also in agreement with prior studies reporting deficient mycorrhization in roots of this mutant plant (*Tejeda-Sartorius, Martínez de la Vega & Délano-Frier, 2008*; *Song et al., 2015*). They differed, however, from

**Table 7 Changes in the abundance of α-tomatine and related compounds in roots at 45 dpi.** The content of α-tomatine, its biosynthetic precursors and catabolic products were determined in roots of wild-type (WT) and spr2 mutant tomato plants after a 32 dpi mycorrhizal colonization period[1].

| Ion identity | Putative metabolite[1] | Wt/spr2 | Wt-M/Wt | spr2-M/spr2 | Wt-M/spr2-M |
|---|---|---|---|---|---|
| mz_413.94 | Dehydrotomatine (tomatidenol) | 1.20 | 0.80 | 0.79 | 1.21 |
| mz_416.89 | Tomatidine and isomers | **2.34**[5] | **0.61**[6] | 1.04 | **1.38** |
| mz_576.89 | ND[2] | **1.69** | 0.67 | 1.00 | 1.14 |
| mz_578.93 | ND | **1.74** | **0.67** | 1.00 | 1.16 |
| mz_916.75 | Hydroxydehydrotomatidine Trihexoside and isomers | **1.61** | **0.50** | 1.17 | 0.69 |
| mz_1004.64 | Tomatidine dihexoside dipentoside and isomers | 1.46 | 0.99 | 1.21 | 1.20 |
| mz_1018.01 | Tomatidine + dihexoside + pentose + deoxyhexose | 1.07 | 0.72 | 0.64 | 1.21 |
| mz_1030.6 | Di-dehydrotomatine | **1.37** | 1.11 | 1.00 | 1.52 |
| mz_1032.75 | Dehydrotomatine and isomers | **1.37** | 0.92 | 1.17 | 1.07 |
| mz_1034.8 | α-Tomatine and isomers | **1.26** | **1.07** | **1.26** | 1.07 |
| mz_1044.78 | UGA[3] 4 | 1.42 | 0.73 | 1.18 | 0.88 |
| mz_1046.84 | UGA 5 | **1.47** | **0.69** | **1.49** | **0.69** |
| mz_1050.7 | Hydroxytomatine and isomers | **1.42** | **0.78** | **1.36** | 0.81 |
| mz_1056.78 | ND | **1.27** | **1.10** | 1.04 | **1.35** |
| mz_1064.77 | Tomatidine tetrahexoside | **1.49** | 0.81 | 1.18 | 1.01 |
| mz_1072.72 | ND | 1.24 | 1.06 | 0.83 | **1.58** |
| mz_1090.77 | Acetoxy-dehydrotomatine | 1.10 | 0.74 | 1.03 | 0.78 |
| mz_1092.67 | Acetoxytomatine | 1.11 | **0.70** | 1.07 | **0.73** |
|  | Total SGA[4]s | **1.32** | 0.97 | **1.19** | 1.07 |

**Notes:**
[1] According to *Itkin et al. (2011)* Plant Cell 23: 4507-4525.
[2] Identity of ion not determined.
[3] Unknown glycoalkaloid.
[4] Total steroid glycolalcaloids.
[5] Numbers in bold in a green background indicate a significantly higher ratio.
[6] Numbers in bold in a orange background indicate a significantly lower ratio.

data obtained from the JA-deficient *defenseless-1* tomato plant mutants (*Howe et al., 1996*) which was found to have increased AMF colonization than WT and prosystemin overexpressing transgenic plants (*Formenti & Rasmann, 2019*). Lower colonization efficiency in *spr2* roots could have partly resulted from their incapacity to significantly increase JA content above control levels in response to AMF colonization (Fig. 2). Since *spr2* mutants are impaired in the orchestration of the JA burst required for the activation of wound- and systemin-related defense responses (*Li et al., 2003*), it may be argued that AMF promote along-term maintenance of constitutively higher JA levels as a strategy to ensure efficient root colonization. This is in accordance with different scales of AMF-associated JA accumulation previously reported in various plant models, including tomato (*Hause et al., 2007*; *Hause & Schaarschmidt, 2009*; *López-Ráez et al., 2010*; *Sánchez-Bel et al., 2016*). A plausible use of this strategy could be to orchestrate a JA-related suppression of immunity responses triggered by microbe-associated molecular patterns in order to enhance a positive plant–microorganism interaction, as previously observed in Arabidopsis roots (*Jacobs et al., 2011*; *Lakshmanan et al., 2012*). Incidentally, increased

JA levels were also in agreement with the proposed priming of JA-dependent defenses in mycorrhizal plants (*Cameron et al., 2013*; *Sánchez-Bel et al., 2016*), a trait that was shown to be abolished in *spr2* mutant plants (*Song et al., 2015*). These scenarios complement other JA-related phenomena proposed to support mycorrhizal colonization (*Hause et al., 2007*; *Tejeda-Sartorius, Martínez de la Vega & Délano-Frier, 2008*).

An additional condition that could have negatively affected AMF colonization in *spr2* plants was the significant SA accumulation observed in mycorrhizal *spr2* roots (Fig. 2). In contrast to a previous report linking SA accumulation with a disrupted ω−3 FATTY ACID DESATURASE7 function in this mutant (*Avila et al., 2012*), SA accumulation above WT levels in *spr2* roots occurred exclusively in response to AMF colonization. Nevertheless, the accumulation of SA in *spr2* roots having deficient AMF colonization was in agreement with copious evidence, in various plant models, linking SA with a negative effect on the mycorrhization process at various stages (*Fernández et al., 2014*; *Liao et al., 2018*). This outcome is considered to be a consequence of the primary role played by SA in plant defense against biotrophic microorganisms (*Pieterse et al., 2009*; *Gutjahr & Paszkowski, 2009*).

Arbuscular mycorrhizal fungi colonization coincided with the strong induction of the *LePT4* mycorrhizal marker (Table 1). Interestingly, the expression levels of this mycorrhizal colonization marker was found to correlate with AMF colonization efficiency only in experiment E1 (Fig. S1), which was the longest (i.e., 50 dpi) and was performed using a single AMF species. Conversely, in E2 and E3, *LePT4* expression levels were inversely correlated with AMF colonization efficiency (Fig. 1; Table 1). The latter combination was in contradiction with findings that established a specific association between this particular phosphate transporter and a functional AMF mycorrhizal symbiosis in *Lotus japonicus*, *Medicago truncatula*, wheat, and rice (*Xu et al., 2007*; *Gutjahr & Parniske, 2013*; *Wang et al., 2017*; *Zhang et al., 2019*). The discrepancy between the results obtained in E1 with those of E2 and E3 suggest that the use of an AMF consortium vs. a single AMF species and, perhaps, the duration of the colonization period could have been contributing factors to the difference observed. Supporting evidence for this possibility is the observation that mycorrhizal phosphate uptake varies among different AMF species. Also, the AMF conditions established in E2 and E3 were more reminiscent of the dynamics of phosphate uptake in the field, which is assumed to be the combined result of the diverse AMF types that contribute separately to phosphate uptake in response to specific environmental conditions, perhaps similar to those encountered in *spr2* roots (*Kobae, 2019*). Other possible explanations for this apparent disparity may be the likelihood that LePT4 could not be essential for the establishment of the mycorrhizal symbiosis in tomato, considering that the expression of other P transporter genes (i.e., *LePT3* and *LePT5*) was also found to be induced by AMF colonization (*Nagy et al., 2005*). In addition, mycorrhizal-specific LjPT4 and MtPT4 phosphate transporters were also regulated by early root responses to phosphate levels in non-mycorrhizal roots (*Volpe et al., 2016*). Another possibility is that mycorrhizal-specific P transporters might have supplementary functions, as supported by the defensive role against pathogenic fungi or abiotic stress recently assigned to mycorrhizal-specific P transporters in wheat (*Zhang et al., 2019*) and tomato (*Volpe et al., 2018*). On the other
hand, the ca. 3-fold reduction in *LeTP4* expression levels that occurred as the colonization period extended from 32 to 45 days, and the complete reversal its expression pattern in mycorrhizal WT and *spr2* roots at 50 dpi (Table 1), was in accordance with the expression of mycorrhizal-specific P transporters in wheat roots, which were found to be induced at different stages of the AMF symbiosis (*Zhang et al., 2019*).

Contrasting AMF colonization in WT and *spr2* roots coincided with the possibility that mycorrhization efficiency was enhanced by the repression of *GAI*. Although this proposal disagrees with the proposed regulatory role played by DELLA repressors of GA signaling during the establishment of AM symbiosis (*Yu et al., 2014*; *McGuiness, Reid & Foo, 2019*) it is, nevertheless, consistent with other studies evoking a more dynamic role for GAs during the colonization process. Thus, GA levels are believed to fluctuate widely in order to maintain a balance suitable for both for DELLA stability and proper AMF colonization (*Foo et al., 2013*; *Martín-Rodríguez et al., 2015*, *2016*).

BR-related genes were strongly repressed in mycorrhizal WT roots, while they were generally up-regulated in mycorrhizal *spr2* roots. This was also in contradiction with the overall positive influence that BRs had in various plant models, including tomato (*Bitterlich et al., 2014a*, *2014b*; *Foo et al., 2016*; *Tofighi et al., 2017*). However, most of this information was obtained using BR-deficient mutants under experimental conditions that may have influenced other plant hormones (e.g., ethylene; *Jiroutova, Oklestkova & Strnad, 2018*), known to affect mycorrhizal colonization. Another possible drawback was that the effect on AMF colonization was revealed only after BR levels were severely reduced (*Foo et al., 2013*). This scenario reflects the paucity of information regarding the precise role played by BRs during the AMF colonization process (*Bedini et al., 2018*; *McGuiness, Reid & Foo, 2019*).

ABA-related gene expression patterns indicated that the positive influence exerted by the mycorrhizal colonization on ABA biosynthesis and signaling (*Herrera-Medina et al., 2007*; *Martín-Rodríguez et al., 2016*), was not affected in the *spr2* mutant. On the other hand, ET gene expression patterns concurred with ample evidence proposing that ET generally has an inhibitory effect on the AMF colonization (*Foo et al., 2013*, *2016*; *Pozo et al., 2015*; *Gomez Monteiro Fracetto, Pereira Peres & Rodriguez Lambais, 2017*). Thus, the majority of the ET biosynthetic and signaling genes analyzed were induced in mycorrhizal *spr2* roots, predominantly at 32 dpi.

Similar to ABA, the positive participation assigned to the 9-LOX oxylipin pathway in the AMF colonization process (*López-Ráez et al., 2010*; *León-Morcillo et al., 2016*) was not affected in the *spr2* mutants. A comparable argument could be used regarding the 13-LOX pathway, where key regulatory genes of the metabolism and regulation of jasmonates such as *LOXD*, *AOS1*, *JMT* and *JAZ2*, previously found to be induced in mycorrhizal tomato plants (*López-Ráez et al., 2010*), had similar expression patterns in both WT and *spr2* mycorrhizal roots. On the other hand, some WR gene expression profiles were contrary to expectancy. Extensively repressed *AROGP* expression in mycorrhizal WT roots was incompatible with the conceived need to partially degrade complex carbohydrates in the extracellular matrix to allow fungal spread and periarbuscular matrix formation (*Liu et al., 2003*). Also, the induction of the *LHA1* gene in *spr2* roots coupled to its

repression in equivalent WT roots, at 32 dpi, appeared to be contrary to the presumed role of H⁺-ATPases as activators of secondary transport systems at the symbiotic interfaces in tomato (*Rosewarne et al., 2007*; *Liu et al., 2016*) and as mediators phosphate transport and plant growth in *M. truncatula* (*Krajinski et al., 2014*). However, this function is apparently performed in tomato by LHA2, a distinct H⁺-ATPase isoform. Thus, *LHA2* transcripts were found to accumulate in mycorrhizal tomato roots, contrary to *LHA1* whose expression was repressed by AMF colonization (*Ferrol et al., 2002*). These workers hypothesized that the selective down-regulation of *LHA1* could reflect a precise role for this transporter during phosphate uptake in tomato, namely in epidermal cells under non-mycorrhizal conditions. They further argued that this resembled the downregulation of two phosphate transporter genes in mycorrhizal *M. truncatula* roots.

The broad repression of *RBOH1* in mycorrhizal WT roots was contrary to the proposed NADPH oxidase-mediated increase of ROS in *M. truncatula* required to facilitate root cortex colonization by AMF arbuscules (*Belmondo et al., 2016a*, *2016b*). The induction of *PS* in *spr2* mycorrhizal roots at 32 dpi was counter to reports showing that mycorrhizal roots of *PS* overexpressing plants yielded significantly higher A% levels than WT roots (*Tejeda-Sartorius, Martínez de la Vega & Délano-Frier, 2008*) and that exogenous systemin promoted pre-symbiotic and early AMF colonization phases in tomato (*De la Noval-Pons et al., 2017a*, *2017b*). A possible argument to explain these apparent contradictions could be that a suppression of certain JA-dependent WR genes is needed to limit JA-related defense responses in order to allow the proper establishment of the mycorrhizal symbiosis (*Gutjahr et al., 2015*; *Martin et al., 2016*). Conversely, the strong repression the *SCP* WR gene in mycorrhizal WT roots could have positively affected AMF colonization efficiency by altering the abundance and activity of secreted proteases, including SCPs that are known to activate defense responses via proteolytic degradation processes (*Kohler et al., 2015*). Apoplastic SCPs are also known to control the generation of peptide signals critical for proper fungal development within the root (*Rech, Heidt & Requena, 2013*). However, the induction of the JA-inducible *PINII* WR marker gene in both WT and *spr2* mycorrhizal roots at 32 dpi was enigmatic.

A ca. 2-fold higher expression level of the *CCD7* gene in mycorrhizal *spr2* roots, compared to equivalent WT roots was observed. Irrespective of this difference, the behavior observed was consistent with most experimental evidence indicating that constant *CCD7* gene activation usually observed during AMF colonization is symptomatic of this enzyme's involvement, together with CCD1, in the production of AMF-induced C13/C14 apocarotenoids. These include α-inolglucoside, the cyclohexenone blumenol and mycorradicin, considered to be signature regulatory metabolites of the AMF symbiosis at late colonization stages (*López-Ráez et al., 2015*; *Hou et al., 2016*; *Lanfranco et al., 2018*; *Fiorilli et al., 2019*). The contrasting *CCD1b* expression levels observed in WT and *spr2* mycorrhizal roots further suggests that apocarotenoid synthesis via CCD1b was differentially regulated in this mutant plant. This finding reinforced the consensus that further research is required to define how apocarotenoid flux is regulated in plants and how these distinctive chemicals, originating from the same metabolic pathway, interact with other phytohormones to regulate the mycorrhization process (*Fiorilli et al., 2019*).

Arbuscular mycorrhizal fungi colonization had no effect on the expression of *PAL5*, while it led to the repression of the *PAL3* gene in mycorrhizal WT roots, contrary to the induction of both *PAL3* and *PAL4* genes in mycorrhizal *spr2* roots at 32 dpi. These results also disagreed with consensual findings indicating the positive role played by PAL enzymes in the AMF colonization process (*Morandi, 1996*), via their crucial function as mediators of the biosynthesis of secondary metabolites that stimulate AMF root colonization (*Mandal, Chakraborty & Dey, 2010*; *Steinkellner et al., 2007*). However, the induced expression of the AMF-colonization responsive *FLS* gene, required for flavonol synthesis (*Scervino et al., 2005*), was unaffected by the *spr2* mutation. This contrasted with the *BEAT* gene, whose expression was ca. 10-fold higher in mycorrhizal WT roots sampled at 50 dpi. This gene could have positively impacted the mycorrhizal process via its negative influence on methyl salicylate synthesis (*Bera, Mukherjee & Mitra, 2017*). Likewise, the ca. 4-fold higher *DXS-2* expression levels detected in mycorrhizal WT roots, was in agreement with the positive role played by this gene in the AMF symbiosis via its key involvement in the MVA isoprenoid biosynthetic pathway leading, among others, to the generation of biosynthetic precursors of the above-mentioned C13 and C14 apocarotenoids (*Walter, Floss & Strack, 2010*; *Floss et al., 2008a*; *Liu et al., 2003*; *Kuhn, Küster & Requena, 2010*).

*CAS1* catalyzes the cyclization of 2, 3-oxidosqualene, a precursor of cycloartenol, a key metabolic intermediary in sterol biosynthesis, whereas *GAME1* codes for a galactosyltransferase involved in the synthesis of steroidal α-tomatine (*Itkin et al., 2011*). Both these genes were induced in *spr2* mycorrhizal roots at 32 dpi, while *CAS1* was widely repressed by AMF colonization in WT roots. A similar expression pattern was produced by the *FPS1*, a gene required for the synthesis of farnesyl diphosphate, an early biosynthetic precursor of sterols and triterpenoids (*Abe, Rohmer & Prestwich, 1993*). These results disagreed, however, with the higher accumulation of tomatine, its biosynthetic precursors and catabolic products in mycorrhizal WT roots, particularly at 32 dpi (Fig. 5; Tables 6 and 7). They were also incompatible with data showing that downregulation of *GAME1* resulted in an almost 50% reduction in α-tomatine levels in tomato leaves (*Itkin et al., 2011*). However, *GAME1* silencing was observed to lead to the upregulation of various genes known to be involved in pathogen defense. It also caused severe morphological alterations due to changes in membrane sterol levels. This scenario suggests that α-tomatine and related compounds may contribute to maintain AMF colonization in tomato by limiting certain plant defense-responses. They may also act as a stabilizing factor during AMF colonization through their contribution to the maintenance of plant cell viability and, possibly, to the enrichment of membrane micro-domains that modify the fluidity and dynamics of the plasma membrane in order to favor symbiotic plant interactions (*Babiychuk et al., 2008*; *Von Sivers et al., 2019*). The above arguments suggest that tomatine and related compounds could be important regulators of AMF symbiosis in tomato. Moreover, tomatine metabolite data obtained in this study coincided with previous reports indicating that tomatine biosynthesis was negatively affected in *spr2* mutant plants (*Montero-Vargas et al., 2018*) and with findings showing that resistance to necrotrophic fungal and oomycete diseases in *spr2* mycorrhizal tomato plants, was

significantly lower than comparable WT plants due to their reliance on the JA-regulated induced systemic response (*Song et al., 2015*). It also raised the possibility that impaired α-tomatine accumulation in mycorrhizal *spr2* roots could have been partly responsible for their lower AMF colonization efficiency.

Another contributing negative factor could have been the induction of SA-related genes, which occurred almost exclusively at 32 dpi in *spr2* mycorrhizal roots (Table 1). Augmented expression of these genes coincided with the significant accumulation of root SA levels and with the induction of *WRKY60*, coding for a tomato TF, closely related to the *Arabidopsi*s WRKY70. This TF is known to regulate the SA-dependent SAR active against biotrophic fungal pathogens (*Bai et al., 2018*), and to inhibit JA-ET-responsive defense gene expression (*Li et al., 2019*). Reduced colonization efficiency also agreed with significantly increased expression levels of the *SAMT* gene, coding an enzyme catalyzing the formation of MeSA, considered to be an important component of the long-distance circuitry needed to establish plant SAR (*Park et al., 2007*; *Shah & Zeier, 2013*). The only discrepancy was the induced expression in mycorrhizal *spr2* roots of the *SSI1* gene, coding for a stearoyl acyl carrier protein FAD that converts stearic acid to oleic acid, a fatty acid known to inhibit SA signaling (*Kachroo et al., 2008*). Perhaps its induction reflected the proposed ability of AMF to modulate SA-related defense signaling. In any case, the reduced mycorrhizal efficiency in *spr2* roots having induced levels of SA-related genes corroborated the above mentioned notion that SA acts as an inhibitor of the mycorrhizal symbiosis (*Blilou, Ocampo & García-Garrido, 1999*; *Herrera-Medina et al., 2003*; *Bedini et al., 2018*).

## Differential fatty acid composition and accumulation patterns in leaves and roots could contribute to the contrasting AMF colonization levels observed in WT and *spr2* plants

A significantly lower content of mostly all FAs except linoleic acid, was detected in mycorrhizal *spr2* leaves and roots. This key biochemical modification could have contributed to the lower mycorrhization efficiency that characterizes *spr2* plants. The increased levels of palmitic acid and a tendency towards higher oleic in mycorrhizal WT roots acid were relevant in the context of the AMF symbiosis. First, augmented palmitic acid in WT mycorrhizal leaves and roots agreed with recent evidence showing that the AM symbiosis increases the lipid flux and redirects it to generate 16:0 β-monoacylglycerols, which are later transferred from the plant to the periarbuscular apoplast (*MacLean, Bravo & Harrison, 2017*). Additionally, higher oleic acid in mycorrhizal WT roots, may have favored mycorrhizal colonization by inhibiting SA signaling (*Kachroo et al., 2004*).

Lower FA content in *spr2* mycorrhizal leaves and roots suggested that the C supply to the mycorrhizal fungi, recently demonstrated to be composed by both lipids and sugars (*Jiang et al., 2017*; *Luginbuehl & Oldroyd, 2017*; *Roth & Paszkowski, 2017*) was probably an additional factor responsible for lower colonization efficiency in *spr2* roots. It remains to be defined if the absence of polyunsaturated 16:3 and 18:3 FAs in *spr2* leaves and roots could have also affected the establishment of AMF by altering the fluidity and/or transport capacity of their plant cell membranes, including the periarbuscular membrane.

This change could have indirectly affected P transport across membranes, or even *spr2* plant fitness, considering that low polyunsaturated FA composition in *fad2* Arabidopsis mutants was shown to reduce the mobility of membrane lipids and to concomitantly impair the $Na^+/H^+$ pump function and the proton translocating activity of ATPases (*Zhang et al., 2012*). Also relevant was the relation to data generated by an untargeted metabolomic analysis in tomato that revealed that α-linolenic acid derivatives were positively affected by the mycorrhizal symbiosis (*Rivero et al., 2015*). Finally, FA data were in accordance with a report showing that, in addition to flavonoids, linolenic acid, and linoleic acid were key factors responsible of regulating AM colonization in litchi trees (*Shu et al., 2016*).

## Differential metabolite accumulation in roots coincided with differential AMF colonization in WT and *spr2* plants

A supervised principal component analysis of signals derived from an untargeted metabolomic analysis of root extracts emphasized the variability observed between mycorrhizal roots of WT and *spr2* plants (Figs. 3 and 4; Tables S3 and S4). Additionally, a GC–MS targeted metabolic analysis revealed that AMF colonization had a much more pronounced effect on metabolite abundance in leaves than in roots of plants sampled at 45 dpi. The relevant increase in sucrose content, coupled to lowered glucose and fructose levels, in WT mycorrhizal leaves, was in agreement with their increased need to redirect sucrose from leaves to roots to support their higher colonization rates compared to *spr2* plants (*Roth & Paszkowski, 2017*). A similar argument could explain the significantly higher level of $PO_4^{3-}$ in leaves of mycorrhizal WT leaves. On the other hand, significantly higher content of DL-malic acid and L-threonic acid in *spr2* roots deficient in mycorrhizal colonization could have mirrored a metabolic condition not amenable to AMF colonization (*Bedini et al., 2018*). With respect to the SA-dependent implementation of the AMF-inhibiting SAR described above, it is germane to add that SA is also believed to involve a repression of fermentation, higher cytosolic oxidative potential and other conditions known to favor AMF colonization (*Bedini et al., 2018*).

## CONCLUSIONS

Root transcriptomic and metabolomic data indicated that reduced AMF colonization in *spr2* roots was likely caused by JA-dependent and JA-independent factors. The latter involved SA accumulation and induction of SA-regulated defense related genes.
The differential expression of key GA and ET genes in mycorrhizal WT and *spr2* roots strongly suggested that alterations in the signaling and/ or biosynthetic pathways of these phytohormones were also central to the regulation of the mycorrhizal symbiosis in tomato. This included the positive association observed between augmented JA levels, induced expression of late JA-dependent WR genes and mycorrhizal efficiency. Targeted and untargeted metabolomic analyses of mycorrhizal WT and *spr2* roots indicated that AMF colonization levels led to significant modifications in their respective metabolomes. These included changes in their fatty acid profiles and in certain metabolites associated with a physiological state favorable to the establishment of the AM symbiosis. In this

regard, the accumulation of tomatine and related SGAs appeared to positively regulate the mycorrhizal process, particularly at earlier colonization stages.

## ABBREVIATIONS

| | |
|---|---|
| **AMF** | arbuscular mycorrhizal fungi |
| **JA** | jasmonic acid |
| **MeJA** | methyl jasmonate |
| **SA** | salicylic acid |
| *spr2* | *suppressor of prosystemin-mediated response2* |

## ACKNOWLEDGEMENTS

We are grateful to Dr. Gregg Howe (Michigan State University) for kindly supplying the *spr2* tomato seeds and the Tomato Genetic Resource Center at the University of California, Davis, for donating the cv. Castlemart tomato seeds.

### Funding

Nicole Dabdoub-González (No. 191369), Josaphat Miguel Montero-Vargas (No. 191369) and Kena Casarrubias-Castillo (No. 227935) were supported by postgraduate scholarships granted by The National Council of Science and Technology (Conacyt, México). Hamlet Avilés-Arnaut was supported by a Basic Science grant conceded by Conacyt (No. 239695). The laboratory for biochemical and instrumental analysis (Robert Winkler) was funded by the CONACyT Fronteras project 2015-2/814 and the bilateral grant Conacyt-DFG 2016/ 277850. The funders had no role in study design, data collection and analysis, decision to publish, or preparation of the manuscript.

### Grant Disclosures

The following grant information was disclosed by the authors:
The National Council of Science and Technology (Conacyt, México): 191369, 191369 and 227935.
Basic Science Grant: 239695.
CONACyT Fronteras Project: 2015-2/814.
The Bilateral Grant Conacyt-DFG: 2016/277850.

### Competing Interests

Robert Winkler is an Academic Editor for PeerJ.

### Author Contributions

- Kena Casarrubias-Castillo conceived and designed the experiments, performed the experiments, analyzed the data, prepared figures and/or tables, authored or reviewed drafts of the paper, and approved the final draft.

- Josaphat M. Montero-Vargas conceived and designed the experiments, performed the experiments, analyzed the data, prepared figures and/or tables, authored or reviewed drafts of the paper, and approved the final draft.
- Nicole Dabdoub-González performed the experiments, analyzed the data, prepared figures and/or tables, and approved the final draft.
- Robert Winkler analyzed the data, authored or reviewed drafts of the paper, and approved the final draft.
- Norma A. Martinez-Gallardo performed the experiments, analyzed the data, prepared figures and/or tables, and approved the final draft.
- Julia Zañudo-Hernández conceived and designed the experiments, performed the experiments, analyzed the data, authored or reviewed drafts of the paper, and approved the final draft.
- Hamlet Aviles-Arnaut conceived and designed the experiments, analyzed the data, authored or reviewed drafts of the paper, and approved the final draft.
- John P Délano-Frier conceived and designed the experiments, analyzed the data, authored or reviewed drafts of the paper, and approved the final draft.

## Data Availability

Data is available at Zenodo:

Kena Casarrubias-Castillo, Josaphat M. Montero-Vargas, Nicole Dabdoub-González, Robert Winkler, Norma A. Martínez-Gallardo, Julia Zañudo Hernández, … John P. Délano-Frier. (2019). Metabolic fingerprints for suboptimal mycorrhizal colonization in wild-type and the jasmonic acid deficient spr2 tomato mutant. Zenodo. DOI 10.5281/zenodo.3560965.

Kena Casarrubias-Castillo, Josaphat Miguel Montero-Vargas, Nicole Dabdoub, Robert Winkler, Norma Martínez-Gallardo, Julia Zañudo-Hernández, … John P. Délano-Frier. (2019). Gene expression for suboptimal mycorrhizal colonization in wild-type and jasmonic acid deficient spr2 tomato mutants. Zenodo. DOI 10.5281/zenodo.3560410.

## Supplemental Information

Supplemental information for this article can be found online at http://dx.doi.org/10.7717/peerj.8888#supplemental-information.

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
