# Peer review of "Distinct gene expression and secondary metabolite profiles in suppressor of prosystemin-mediated responses2 (spr2) tomato mutants having impaired mycorrhizal colonization"

_PeerJ, doi:10.7717/peerj.8888_

## Round 0.1 · original submission · Major Revisions

Dear authors,

First of all I want to apologize for the long time period the evaluation of your manuscript needed, but it was rather difficult to find experts to be willing to provide a reviewer report.

As you can see from the reports, all three reviewers were not very satisfied with your manuscript and the data, which are presented. Most importantly, all three reviewers highlighted the discrepancy between the mycorrhization rate determined by staining and the transcript and metabolic data. This might be mainly due to the fact that the current way of presenting these data in context to colonization (based upon staining data only) leads to incorrect conclusions. Here, re-evaluating the data using AMF marker genes is urgently needed.

If you want to go for a revision of your manuscript, you should follow all the points raised by the reviewers. In particular, it is urgently needed that you address the following major points:

- Reproducibility of your data by introducing new experiments;
- Statements limited to significant results only;
- Inclusion of data obtained from non-mycorrhizal plants (wild type and spr2) (urgently needed);
- Quantification of hormone levels, at least JA;
- Inclusion of physiological parameters such as plant growth, mycorrhizal dependency, fitness, etc.;
- Extensive rewriting of the discussion.

Reviewer 1 ·

Basic reporting

The authors report about a comparison between wild type and spr2 (JA biosynthesis) mutant tomato plants with respect to colonization by mycorrhizal fungi after about seven weeks of colonization. They analyzed an extremely wide variety of parameters including a wide range of transcripts but also many metabolites (many with unknown identity) across various biosynthetic pathways. The ms is nicely written and presents its content in a good english but the mere wealth of information and the many ups and downs of transcripts and metabolites (some significant, some not) makes it very difficult to find out any truly significant and meaningful results. The figures and tables are clear and nicely presented including appropriate statistis in most parts but again it is very difficult to distill relevant information from the huge amount of data presented, much of it having non-significant or contradictory alterations, whose reproducibility is doubtful.

Experimental design

One point that adds to this enormous complexity and complicates conclusions very much is the use of three different colonization levels defined by microscopic analysis of mycorrhizal structures originating from three independent experiments (Fig. 1). The authors themselves admit that they are unaware what caused these variations. Perhaps it is in part due to the different seasons when these experiments/plant cultivations were performed but the authors do not assign a distinct cultivation season to a distinct colonization level yet this information should perhaps be added.

Validity of the findings

A major problem with this experimental design is that the three colonization levels are not that clearly distinct in all aspects and that non-mycorrhizal controls are shown only in some cases. The „high colonization“ is characterized only by high arbuscule frequency (Fig. 1) whereas frequency of colonization (F%) is not different from „medium colonization“ and only slightly above „low colonization“. Most importantly, many conclusions are made for specific colonization levels only (e.g. even in the title) and this is very problematic when looking at the rather minor differences between colonization levels in some cases (all regarding differences in wild type genotype, spr2 background has no significant differences at all). I have strong doubts that these results will be reproducible in future experiments. The whole concept of comparing different colonization levels can only work if there are really clear differences between such levels. With the current variability between colonization levels I would limit the analysis to one (strong oder medium) colonization level and compare it to the non-mycorrhizal state (which is actually missing in many but not all data displays of the current ms, better have it uniform). Some of the molecular marker data (e.g. LePT4 that I also see as a reliable marker for a mycorrhiza with functional arbuscules) are in at least partial contradiction (for medium colonization) with the colonization levels defined by the microscopic analyses. Also, some genes involved in specific biosynthetic pathways appear to be regulated in a strange manner, e.g. repression of CCD7 and CCD1 during high colonization (Tab. 1) and high arbuscule abundance (high LePT4 (Tab. 1) , high arbuscule score by Mycocalc (Fig. 1)) - makes no sense at all. Also, the glycosyl transferase CGT1 (highly uncertain if involved in pathway) is not induced in high colonization roots. Moreover, some of the data obtained from leaves are also very strange, e.g. that LePT4 is induced in leaves – contrary to all previous reports. The only rather clear conclusion seems to be that JA biosynthesis-deficient plants are moderately compromized in mycorrhizal colonization (not a new finding). Moreover, the reduction of tomatine in the strongly colonized state could be interesting but this is then not represented in the title.

Additional comments

The ms adds to rather than solves the current confusion of whether JA is relevant for mycorrhization and if so whether it is acting positively or negatively and in what stages of mycorrhiza development. The ms lacks a clear focus. Already the title is very difficult to understand and misleading and it is not clear why one should have an interest in what is going on during „suboptimal“ mycorrhizal colonization and why a „suboptimal“ state should be important. In many cases results are presented as „tended to be significant“ or similar. Statements should be limited to clearly significant changes with very few exceptions. These problems are not only due to the use of the different colonization levels, which in my judgements (as discussed above) are not distinct enough, but this concept adds to the problems of the ms. A simple comparison of non-mycorrhizal vs mycorrhizal states for the two genotypes throughout the ms would probably have been the better choice. The discussion is very knowledgeable and phrased nicely but is much too long.

MINOR POINTS
The review of Gutjahr, 2014 does not cover JA. Other references are mostly adequate.

Reviewer 2 ·

Basic reporting

The manuscript by Casarrubias-Castillo and co-workers aims to deepen in the role of JA in the regulation of the AM symbiosis. For that, they analyze the transcriptional and metabolic profiles of the JA-deficient mutant spr2 compared with its corresponding wild-type accession, and at different colonization levels (‘low’, ‘medium’ and ‘high’). The topic is timely and interesting. The authors have done a lot of work, and presented a manuscript with a good and updated Introduction, and well described Materials and Methods. However, probably because they have tried to present too much data, Results and Discussion sections are a bit messy, making the reading and comprehension difficult for readers. It must be more focused. Moreover, the paper is not compensated, as the Discussion is about 50% of the whole manuscript. This section should be reduced and be more focused.

Experimental design

Experiments were well design and, as I mentioned before, Material & Methods are well described. Just one remark, in order to obtain 3 different levels of mycorrhization, instead doing 3 independent experiments, it might be better to performe a time course assay and repeat it twice.

Validity of the findings

As I said, Results and Discussion sections are a bit messy, which difficults the reading and comprehension of the manuscript. In my opinion they should be more focused, specially the Discussion. Moreover, maybe because auhtors wanted to present so many data and information, some conclusions are not well stated. In some cases, marker genes do not behave as expected. I suggest removing the 'medium' mycorrhization level experiment to get rid of the background noise. By doing so, I think data will be more reliable.

Additional comments

The manuscript by Casarrubias-Castillo and co-workers aims to deepen in the role of JA in the regulation of the AM symbiosis. For that, they analyze the transcriptional and metabolic profiles of the JA-deficient mutant spr2 compared with its corresponding wild-type accession, and at different colonization levels (‘low’, ‘medium’ and ‘high’). The topic is timely and interesting. The authors have done a lot of work, and presented a manuscript with a good and updated Introduction, and well described Materials and Methods. However, probably because they have tried to present too much data, Results and Discussion sections are a bit messy, making the reading and comprehension difficult for readers. It must be more focused. Moreover, the paper is not compensated, as the Discussion is about 50% of the whole manuscript. This section should be reduced and be more focused.

As mentioned, my main concern is about the huge amount of data presented, where some of them are not relevant for the main goal. Thus, tables with transcriptomic and metabolic data could be reduced in order to show those that are key for the paper. The whole set of data could be presented as supplemental. Many times ‘less is more’! In the same line, I think that data showing the ‘medium’ colonization levels could be removed, because its mycorrhizal levels are either equal to the ‘low’ or to the ‘high’ samples (see Fig 1). In any case, the two latter are the most contrasting and they would be sufficient for comparison. Moreover, some of the ‘rare’ results come from the analysis of the ‘medium’ samples. Thus, I think that removing them, authors could get clearer conclusions.

The results about the 9-LOX and 13-LOX pathways are very strange. spr2 encodes for a ω3 fatty acid desaturase FAD7, thus blocking not only the 13-LOX pathway producing jasmonates, but also de 9-LOX giving rise to other oxylipins. This fact must be considered, as other JA-deficient mutants are not altered in mycorrhization. On the other hand, even though spr2 is defined as a JA-deficient mutant, I suggest quantifying hormone levels, at least JA, since they are crucial to understand the changes observed along mycorrhization.

Physiological parameters such as plant growth, mycorrhizal dependency, fitness, etc should be included in the manuscript in order to get an idea about the phenological stage of the plants.

Transcriptional results with different marker genes are strange. For instance, DXS2, a well characterized mycorrhizal marker, here is not induced in myc plants. Regarding the metabolic data, the authors could present those that change, independently of the p value (0.05, 0.01 or 0.001). They might also group these compounds by families better than by m/z. This would help to make the results clearer.


Minor points

- Figure legends are poorly described

- Some marker genes used are not properly described in the text. Please explain what the different marker genes are, indicating the corresponding references.

- Line 267, ‘AMF colonization levels coincided with expression of LePT4’. According to Fig.1 and Table 1, this is not completely true. The same happens for the gene CCD7. Please check and correct.

- Instead presenting the results of the a-tomatine as tables, I suggest showing them in a figure with the biosynthetic/catabolic pathway, indicating which compounds are up or down. This would help to globally visualize how this pathway is altered by AM symbiosis.

Reviewer 3 ·

Basic reporting

No comments

Experimental design

Please explain why non-mycorrhizal Wt and spr2 roots or shoots haven’t been considered for the gene expression analyses.

Validity of the findings

The authors analyzed gene expression and metabolite profiles of mycorrhizal wild-type and jasmonic acid deficient (spr2) tomato mutants. Colonization levels were determined by a non-vital staining technique (trypan blue staining) and by transcript analyses of one plant (LePT4) and 2 fungal marker genes (18S Gi and EF1-α Gi). The authors used the staining data to classify experiments as yielding “low”, “medium” and “high” colonization and discussed transcript and metabolite data in context of this classification.

Remarkably, the transcript levels of the AM marker genes don’t correlate with the results obtained from the trypan blue staining. How do the authors explain the opposite results obtained with trypan blue staining and qPCR? Could the authors include more marker genes to validate the findings obtained with trypan blue?

Some of the expression data presented in context of the three colonization levels are not in line with previously published data, e.g. CCD1 transcript levels in colonized wild-type plants decrease when colonization levels are higher (line 297). These can be indications that the staining data don’t reflect the colonization status in the part of the root system used for transcript and metabolite analyses (root system was divided lengthwise during harvest). I am highly concerned that current way of presenting the data is misleading. I suggest presenting transcript and metabolite data in context to colonization levels classified based upon AMF-marker genes which reflect the colonization level of the material used for these analyses.

Additional comments

The authors provide an extensive set on transcript and metabolite data from wild-type and spr2 mutant plants that may lead to additional findings in how JA signaling affects AM symbiosis if presented in the correct way.

---

## Round 0.2 · Minor Revisions

Dear authors,

Thank you for submitting your revised manuscript to PeerJ.
The manuscript has been evaluated by two reviewers, who already evaluated the former version. As you can see from the reports, both reviewers found the manuscript in general suitable for publication in PeerJ, but expressed concerns that require your attention. Both of them explain in detail what in their opinion would be necessary to improve the manuscript and make it suitable for publication in PeerJ.

To perform a revision of your manuscript, you should follow all the points raised by the reviewers. In particular, it is urgently needed that you address the following points:
• Presentation of the expression data for each genotype (WT and spr2) and treatment (non-mycorrhizal and mycorrhizal) separately by calculating gene expression relative to the reference gene (2ΔCt). This will allow a comparison between both genotypes as requested by both reviewers;
• Inclusion of expression levels of other AMF marker genes to support the main findings;
• Adding 50-dpi-columns for WT and spr2 in Table 1.

I’m looking forward to get the revised version of your manuscript.
Best regards
Bettina Hause

Reviewer 2 ·

Basic reporting

In this new version, the authors have addressed all or most of the reviewer’s suggestions and introduced changes accordingly. They have removed some of the previous experiments that were ‘controversial’ and included 2 new mycorrhizal experiments. The MS has been clearly improved. It reads better, and the results, conclusions and final message are much clearer now.

Experimental design

As mentioned, 'strange' data from previous experiments were removed. With the inclusion of 2 new experiments the results are more focused and clear now.

Validity of the findings

Results from 3 independent experiments are represented, so the results are consistent and the conclusions well stated. Only one remark, although the authors made an effort to explain the discrepancy with the results regarding the mycorrhizal marker gene LePT4, they fail to find a 'feasible hypotesis'.

Additional comments

This is a new version of previous MS PeerJ-34877, which has been surely improved. The authors have addressed all or most of the reviewer’s suggestions and introduced changes accordingly. They’ve eliminated some of the previous experiments that were ‘controversial’ and included new mycorrhizal experiments and analyses. The MS reads better, and the results and message are much clearer now.

I only have a few remarks, according to the ‘physiological’ data, spr2 is smaller, and it has less chlorophyll content, etc. The same applies to transcriptional data. The comparison between the two genotypes is not made, and I think is important. On the other hand, it should be nice to analyse the expression of all set of genes in the same samples. Thus, you’ll have a better comparison.

The authors made an effort to explain the discrepancy with the results of the marker LePT4. This gene is also a good mycorrhizal marker in tomato, and there are several papers that prove it.


Minor points:

- In line 158, in addition to the final concentration of P used (7 µM) indicate the ‘final’ percentage to get an idea of the reduction.

- In M&M when describing the qRT-PCR analyses, instead using 3 technical replicates of a pool from 3 biological samples, it should be better to analyse the biological replicates per separate and then calculate the SE.

- I’d include the Trouvelot data from E1, at least as supplemental, not just in the text.

- Since the MS is mainly based on the changes produced in roots, when analysing the metabolic profiles I’d start by the roots instead the leaves.

- In Fig. 3, the letters a, b, c and d are not present.

Reviewer 3 ·

Basic reporting

Casarrubias-Castillo completely revised their manuscript which aims to explain reduced AMF colonization in spr2 mutants by analyzing transcript and metabolite data. Instead of using three different colonization levels for the interpretation of their data, the authors performed two additional experiments that mirrored the high colonization levels from the first experiment and included these data in the revised version. Additionally, relevant hormone measurements were included (JA and SA). I doubt that the addition of targeted metabolite analyses in roots and leaves (table 2 & 3) really contributes to the aim of the paper. In fact, the discussion of some of these data is very speculative (discussion section lines 734 – 742) and can be removed.
The manuscript is nicely written. However, some parts of the discussion are very speculative and can be shortened. One example was already mentioned, another one is the discrepancy between mycorrhizal colonization and LePT4 expression which will be explained in the “comments to the author’ section.
The figures are relevant. The placing of the lower case letters in figure legends 2, 3 and 5 is confusing. Additionally, figure legend 3 refers to lower case letters (a-d) that are missing in the figure.

Experimental design

As already mentioned, the authors performed two more experiments to validate their data (E2 and E3). For these two studies, an AMF consortium consisting of six AMF species was used. From the Material and Methods part it is not clear if these two additional experiments were run in parallel and started at the same day, or setup at different time points. This information may be useful for the interpretation of some data.

Validity of the findings

Colonization data of these two additional experiments align very well with an experiment performed with only one single AMF species (E1, 50dpi). However, LePT4 expression data for E2 and E3 don’t align with colonization data obtained from fungal staining. The current MS includes expression data only for selected genes where E1 data were added instead of 45dpi expression data. It would be helpful if the authors include all gene expression data from E1 to table 1 to show the reproducibility of the data. This can be easily addressed by adding 50dpi columns for WT and spr2 in Table 1.

Additional comments

LePT4 expression and expression of other genes known to be induced during AMF colonization (e.g. CCD7, LHA1) keeps looking strange and I am not satisfied with the discussion of this discrepancy. As outlined by the authors in their comments to the reviewer, the current way of presenting the expression data reflects the expression relative to the non-mycorrhizal plants (2-ΔΔCt method). I am wondering if useful information, e.g. expression of genes in non-mycorrhizal roots of spr2 plants, is lost when this method is used. I suggest presenting the expression data for each genotype (WT and spr2) and treatment (non-mycorrhizal and mycorrhizal) separately by calculating gene expression relative to the reference gene (2ΔCt). Maybe some genes are differentially regulated in non-mycorrhizal spr2 roots? This finding may open a more reliable avenue for interpretation. The current way of discussing the discrepancy between fungal staining and LePT4 expression is very speculative.
The authors report testing of other AMF marker genes to support their findings in their comments to the reviewers. Why can’t these data be included in this MS?

---

## Round 0.3 · Minor Revisions

Dear authors,

Thank you for submitting your revised manuscript to PeerJ. You have properly addressed all the concerns raised by the reviewers and by me regarding the former version. There is only one point you should address before the manuscript can be finally accepted (as identified by the Section Editor) :

The real-time RT-PCR approach you used is well suited for validation of differential expression, but reference sequence within the genome needs to be highlighted so that the reader can extract the sequence to test to potential gene diversity in the genome. Therefore, it is required to add the respective Solyc number of each gene tested. This might be done easiest by introducing the numbers into the table showing the primers used (Table S1).

Best regards
Bettina Hause

---

## Round 0.4 · accepted · Accept

Dear authors,

it is my great pleasure to finally accept your manuscript for publication in PeerJ! You were right, it was a long-lasting story, but now it finished successfully. Also from my side: Congratulations!

Best regards
Bettina Hause